# Implications of Metastable Nicks and Nicked Holliday Junctions in Processing Joint Molecules in Mitosis and Meiosis

**DOI:** 10.3390/genes11121498

**Published:** 2020-12-12

**Authors:** Félix Machín

**Affiliations:** 1Unidad de Investigación, Hospital Universitario Nuestra Señora de Candelaria, 38010 Santa Cruz de Tenerife, Spain; fmachin@fciisc.es; 2Instituto de Tecnologías Biomédicas, Universidad de la Laguna, 38200 Tenerife, Spain; 3Facultad de Ciencias de la Salud, Universidad Fernando Pessoa Canarias, 35450 Las Palmas de Gran Canaria, Spain

**Keywords:** holliday junction, homologous recombination, Mus81, GEN1/Yen1, BLM/Sgs1, Mlh1-Mlh3/MutLγ, replication stress, double strand breaks, dissolution pathway, ZMM pathway

## Abstract

Joint molecules (JMs) are intermediates of homologous recombination (HR). JMs rejoin sister or homolog chromosomes and must be removed timely to allow segregation in anaphase. Current models pinpoint Holliday junctions (HJs) as a central JM. The canonical HJ (cHJ) is a four-way DNA that needs specialized nucleases, a.k.a. resolvases, to resolve into two DNA molecules. Alternatively, a helicase–topoisomerase complex can deal with pairs of cHJs in the dissolution pathway. Aside from cHJs, HJs with a nick at the junction (nicked HJ; nHJ) can be found in vivo and are extremely good substrates for resolvases in vitro. Despite these findings, nHJs have been neglected as intermediates in HR models. Here, I present a conceptual study on the implications of nicks and nHJs in the final steps of HR. I address this from a biophysical, biochemical, topological, and genetic point of view. My conclusion is that they ease the elimination of JMs while giving genetic directionality to the final products. Additionally, I present an alternative view of the dissolution pathway since the nHJ that results from the second end capture predicts a cross-join isomerization. Finally, I propose that this isomerization nicely explains the strict crossover preference observed in synaptonemal-stabilized JMs in meiosis.

## 1. Introduction

Homologous recombination (HR) comprises a complex mechanism, whereby it is possible to anneal homologous sequences in the genome. HR serves two purposes: (i) to combine genetic information and faithfully segregate homolog chromosomes in meiosis and (ii) to repair DNA damage in mitosis. The first HR steps are relatively identical in both situations, but the last steps differ since it is of the interest of meiosis to exchange genetic information, whereas during DNA repair this exchange must be kept to the minimum. HR is extremely beneficial for the somatic cell, the organism, and the species. HR keeps cells alive and genetically stable since it is the most error-free mechanism to deal with DNA damage. Likewise, multicellular organisms benefit from maintaining their cells genetically stable as it suppresses death-threatening diseases such as cancer. As for the species, HR in meiosis promotes genetic diversity of the offspring.

Even though HR is clearly beneficial, it puts cells at risk in the short-term. HR brings about linkages between sister or homologous chromosomes that can interfere with their segregation in anaphase. These linkages are generally referred to as joint molecules (JMs), and there are basically two types: the displacement loop (D-loop) and the Holliday junction (HJ). Throughout the years, many studies have identified proteins and mechanisms devoted to eliminating these JMs before chromosome segregation ([1,2,3] for recent reviews). The proteins involved in dealing with JMs are highly conserved from bacteria to humans and gather around three classical enzymatic groups: helicases, type IA topoisomerases, and structure-specific endonucleases (SSEs). Among the former two, one helicase-topoisomerase complex stands out: the RTR complex (BTR in human; STR in *Saccharomyces cerevisiae*, etc.), which is formed by a RecQ-like helicase (RecQ in *Escherichia coli*, BLM in humans, Sgs1 in *S. cerevisiae*, Rqh1 in *Schizosaccharomyces pombe*, etc.), Topoisomerase 3 (Top3 in both yeasts; TOPOIIIα in humans), and a cofactor (Rmi1; RMI1-RMI2 in humans). Regarding SSEs, three proteins have been shown to cut JMs in vitro: the heterodimers MUS81-EME1 (Mus81-Mms4 in *S. cerevisiae*) and BTBD12/SLX4-SLX1 (Slx4-Slx1), and the homodimer GEN1 (Yen1). Genetic studies have found that the Mus81/MUS81 heterodimer (referred to as Mus81* hereafter) and GEN1/Yen1 eliminate JMs that emerge from HR, whereas the role of Slx4-Slx1 is more complex and not conserved [4].

The importance that timely removal of JMs has for the immediate cell fate is highlighted by the fact that mutants for RTR and SSEs present anaphase bridges [5,6,7,8]. Conceptually, this is not surprising since the presence of just a single HJ should compromise segregation (Figure 1A). Anaphase bridges are a source of genome instability with predicted outcomes that include chromosome number variation and terminal loss of heterozigosity (tLOH) [9,10,11]. The coordinated actions of RTR and SSEs serve the purpose of removing JMs, but they do not act on the same JMs, nor do they necessarily work at the same time. Recent literature in *S. cerevisiae* has shed light on the latter and established a temporal hierarchy: RTR in S-phase, Mus81* in G2/M and GEN1/Yen1 in anaphase (Figure 1B) [12,13,14,15,16]. As for the specificity, The RTR complex eliminates HJs in couples (double HJ; dHJ) in the so-called dissolution pathway [17,18]. On the other hand, each SSE owns specificity for distinct JMs and cuts at different points of the branched DNA [19,20,21]. SSEs actions on HJs encompass the so-called resolution pathway, and these SSEs are frequently referred to as resolvases.

The most important eukaryotic SSE appears to be Mus81*, with GEN1/Yen1 acting as a backup. Surprisingly, Mus81* is per se a poor resolvase against the canonical HJ (cHJ), yet an excellent nicked HJ (nHJ) resolvase [20,22]. The cHJ is continuous four-way branched DNA structure, and it has been central in all HR models. The nHJ is a HJ with a single strand break (a “nick”) at the branch point. The nHJ has been neglected as a real intermediate in most HR models, despite it is often a necessary precursor. We showed in the past that yeast cells depleted from Mus81* and Yen1 formed anaphase bridges largely enriched in non-canonical HJs [6]. At least 50% of these were nHJs. In this conceptual paper, I explore the effects of having nHJs as metastable intermediates in HR models that deal with double strand breaks (DSBs) and replication stress (RS). Moreover, I also explore the effects of nicks in the vicinity of HJs. Recent literature underlines the importance of such nicks during the resolution of HJs in meiosis by the endonuclease MutLγ (MLH1-MLH3; Mlh1-Mlh3 in yeast) [23,24,25].

## 2. Biophysical, Biochemical, Topological and Genetic Features of cHJs and nHJs

Before presenting the origin and fate of JMs in the context of the HR pathway, I start with the theoretical implications of having cHJs and nHJs. I discuss this from a thermodynamic, topological, enzymatic, and genetic point of view.

### 2.1. A Single Canonical HJ (cHJ)

The cHJ is a continuous covalently-bound four-way JM. In HR schematics, the cHJ is frequently depicted in parallel with one of the two homologous molecules vertically flipped, so that the two crossing strands can be easily visualized on the plane (Figure 2A). DNA strands are often depicted as straight lines for simplification, although this is far from representing the double helix of B-form DNA. An alternative representation can be drawn to better reflect this fact (Figure 2A, right). Although the parallel conformation is the easiest to follow in HR models, within the cell the actual conformation is unknown. The cHJ can acquire many three-dimensional shapes. Aside from the parallel conformer, two other stereoisomers are especially important: the stacked antiparallel conformer and the open cruxiform conformer (Figure 2B) [26,27]. The antiparallel conformer results from rotating 180° one of the homologous in Z (XY coordinates define the plane of the schematic). The open conformer results from rotating 90° in the XY plane two of the four arms in the antiparallel conformer. The parallel and antiparallel conformers own two-fold symmetry, with just two strands making the crossing overs at the cross-join (reddish lines in isomer 1). On the contrary, the open conformer owns 4-fold symmetry, with all four strands equally involved in the junction. Transitions out of the open conformer back into the antiparallel forms can lead to an important isomerization reaction whereby crossing-over strands interchange (Figure 2B; isomer 2).

Biophysical data obtained with short “immobile” cHJs (designed to avoid branch migration) suggest that the parallel conformation in rather unfavorable, likely because of strong steric and electrostatic repulsion among the phosphate backbones [26,27]. Instead, a stacked antiparallel conformer, with the arms tilted 60° in Z relative to one another, seems to be preferred when divalent cations compensate for the negatively-charged phosphate repulsion (Figure 2C). This stacked antiparallel conformation only occurs in one of the two possible isoforms in vitro, and spontaneous crossing-over isomerization has not been reported. 

Another important hallmark of the cHJ relates to the topology of the cross-join [28]. I highlight two topoisomers, which differ in the relative over or under position of the cross-over strands (Figure 2D). These topoisomers may have profound implication in the outcome of molecular rotations and, perhaps, in the dissolution pathway (see below). Regarding the molecular rotation, the parallel/antiparallel conformational isomerism depicted in Figure 2B would need a specific one-direction rotation for topoisomer 1 and the opposite for topoisomer 2. Otherwise, rotation would result in braiding of the cross-over strands [29], with uncertain results.

Even though these biophysical features of cHJ are well known, it is remarkable that they are mostly overlooked in HR models. It is difficult to determine their importance in vivo within the constrained space of the nucleus/nucleoid, amid the tangled chromatid mess. For instance, assuming that cHJ formation occurs in a parallel conformation, cross-over strand isomerization would require conformational transitions that would bend a large portion of a chromosome. On the other hand, what seems topologically and thermodynamically unlikely is confronted by the fact that cells possess topoisomerases and isomerases that drive kinetically unfavorable reactions at the expense of ATP. In any case, an important message is that the parallel conformation is thermodynamically adverse in cHJs; therefore, cells either actively maintain parallel cHJs or actually have cHJs in alternative conformations. Interestingly, several proteins that recognize and metabolize cHJs prefer/stabilize the open conformation [30,31,32].

A key characteristic of the cHJ, which is indeed considered in HR models, is that cHJs can branch migrate. Branch migration is in fact an isomerization reaction whereby the cross-join point moves along the homologous molecules (Figure 2E). Branch migration cannot take place if the cHJ is in the stacked antiparallel conformation [33,34]. However, branch migration occurs in vitro very rapidly through a random walk mechanism in unstacked conformations, especially when topological constrains are not present (e.g., all four arms can freely rotate). In vivo, branch migration appears to be governed by helicases and translocases, expending ATP in the process [30,35,36]. Part of this cost in energy may account for the need of conformational changes; from early works in the *Escherichia coli* 5′→3′ helicase RuvAB, it was determined that the open conformation was the one suitable for branch migration [37]. Additionally, branch migration must not follow a random walk but should have directionality in many instances—e.g., during dissolution of dHJs. Lastly, branch migration of a single cHJ delimited by topological barriers for its four arms gives rise to over-winding (positive supercoiling; (+)SC) ahead and under-winding (negative supercoiling; (−)SC) behind (Figure 2E). Thus, in most cases, it is expected that helicases/translocases propel branch migration, with topoisomerases dissipating (−)SC and (+)SC. Aside from prokaryotic helicases (e.g., RuvAB), several eukaryotic helicases/translocases belonging to the SF2 3′→5′ helicase superfamily and tightly associated with HR have been shown to branch migrate cHJs in vitro [36]. These include RecQ-like (Sgs1, Rqh1, BLM), FANCM (Mph1 in *S. cerevisiae*), SRS2/ScSrs2, HTLF/ScRad5, and RAD54/ScRad54. On the other hand, both type I and type II topoisomerases (Top2) could deal with HJ-mediated supercoiling. The actual roles of type IB topoisomerase (Top1) and Top2 are still a matter of controversy [38,39]. In principle, they could both relax supercoiling; Top1 makes a transient nick that enables strand rotation until the torsional stress is dissipated, whereas Top2 makes a transient DSB so that another portion of the DNA molecule is transported through with similar relaxation effects. A third topoisomerase, type IA TOPOIIIα/Top3, has been more closely implicated in branch migration owing to its above-described association with RecQ helicases (eukaryotic RTR complexes). Because RTR can efficiently branch migrate HJs, it is believed that RecQ helicases are the basic cHJ propeller in vivo, with Top3 dissipating supercoiling [17,40,41]. Top3 makes single-stranded DNA (ssDNA) incisions and then passes another ssDNA through it. In this way, Top3 acts as a ssDNA decatenase/catenase, so that its putative role in relaxing supercoiling must be clearly distinct from Top1 and Top2 and is not precisely understood.

Lastly, the cHJ is the target of specialized SSEs—i.e., HJ resolvases. Bacterial RuvC, and eukaryotic GEN1/Yen1 are the archetypical resolvases. They form homodimers over the cHJ that resolve the JM through a double counter-nicking mechanism [32]. The endonucleolytic resolution of a cHJ can lead to two outcomes owing to the symmetry of the JM (Figure 2F; “a” vs. “b”). RuvC and GEN1/Yen1 work on the 4-fold symmetric open conformer, so the chances of cutting through either plane are theoretically equal. These distinct outcomes can have significant genetic consequences in HR models (see below). When resolution of a single cHJ is considered in isolation, the different resolution planes just affect the strands whose crossover are fixed. If the cHJ connected two homologous chromosomes, rather than sister chromatids, the exchange of strands creates a tract of heteroduplex DNA (hxDNA). This tract may contain sequence dimorphisms that must be resolved by the mismatch repair (MMR) machinery, leading to the conversion of each dimorphism into just one variant [42]. This, in turn, results in either restoration of the original sequence or transposition of sequences between the recombinant molecules. Sequence transposition, which can be unidirectional, bidirectional, uniform, or patchy, is one of the two origins of gene conversion (GC) events observed in HR (the other one being de novo DNA synthesis; see below). Of note, the concepts of hxDNA and GC only make sense in the context of JMs connecting two homolog chromosomes. However, I use them here indistinctly for homolog and sister chromosomes. The equivalence of hxDNA tracts between sister chromatids are tracts of seemingly conservative replication—i.e., double-stranded DNA (dsDNA) comprising either parental or daughter strands, flanked by the predicted semiconservative dsDNA formed by one parental and one daughter strand.

### 2.2. A Single Nicked HJ (nHJ)

The nHJ, a four-way JM with a nick at the junction (Figure 3A), is also a preferred in vitro substrate for RuvC and GEN1/Yen1 resolvases [15,43,44]. Most importantly, it is a preferred substrate of Mus81* [20]. As mentioned before, Mus81* is generally more important than GEN1/Yen1 in HR [45,46,47]. These facts should make the nHJ an intermediate to be seriously considered in all HR models.

The nHJ lacks many of the thermodynamic and topological constrains of the cHJ. The nick at the junction places the molecule in a local minimum from a thermodynamic point of view. The strands are more freely to rotate and search for minimum electrostatic and steric interactions among them. In fact, the open conformer seems to be preferred in all in vitro conditions (Figure 3B) [48]. There is just one possible isomer, the one with just one cross-over strand. Thus, the common drawing of the nick onto one of the strands which does not participate in the crossover is misleading. Moreover, the topological concern about the relative disposition of the strands in the cross-join is absent in the nHJ.

Because of the presence of a preexisting nick, resolution of nHJ by a counter-nicking mechanism is directional (Figure 3C). The action of RuvC and GEN1/Yen1 results in just one of the two possibilities described for cHJs. The case of Mus81* is similar, albeit the cut occurs a few nucleotides away from the branch point. This leads to one product with a 5′ flap and the second product with a small ssDNA gap. These modifications are easy to deal with in vivo to end up with two continuous dsDNA molecules. Interestingly, this asymmetric counter-nicking action performed by Mus81* may give separate identities to the resulting products.

The nHJ is probably metastable as it occupies a local thermodynamic minimum. However, isomerization into a cHJ through branch migration is possible in solution [49], and it is, therefore, likely in vivo as well (spontaneously or enzymatically driven). Interestingly, the nick that is left behind after branch migration should ameliorate the need for the concerted action of topoisomerases as they enable strand swiveling during branch migration (Figure 3D).

### 2.3. A Partner of Canonical HJs: The Canonical Double HJ (dcHJ)

The combination of two HJs deserves separate chapters due to (i) their emerging properties, and (ii) most DSB repair and RS bypass models predict two HJs. I start with the dHJ formed by two cHJs (dcHJ)—i.e., the canonical dHJ.

The first important fact when considering dcHJs is that they could come in two basic isomers in the parallel configuration. In the first isomer, both cHJs have the same cross-over strands, so that each strand returns to its parental molecule once it has crossed the dcHJ (Figure 4A). This kind of isomer always has an even number of helical half-turns between the cHJs [27]; it is the one considered in HR models and is formally named as DPE (D, double; P, parallel; E, even). The second isomer comprises two cHJs with opposing cross-over strands (Figure 4B). In this case, an odd number of helical half-turns is found between the cHJs, and it is formally named as DPO (D, double; P, parallel; O, odd) [27]. Conformational isomerization is also predicted for the dcHJ. In fact, several “immobile” dcHJ conformers can be designed in vitro, with a clear stability preference for antiparallel conformers when there are no topological constrains [50]. This highlights the unfavorable thermodynamics of dcHJs in parallel conformations. However, in vivo conformational isomerization to antiparallel conformers is challenging and counterintuitive for whole chromosomes. Likewise, isomerization of crossing strands for individual cHJs seems challenging, at least without the aid of topoisomerases.

Branch migration is biologically relevant in dcHJ and can only take place in parallel conformers [50]. In topologically constrains dcHJs, converging migration generates twice (+)SC/(−)SC than a single cHJ that travels the same distance (Figure 4C). (+)SC between the converging cHJs is a major issue when they get very close to one another. There must be a limit in the distance between cHJs that cannot be surpassed without distorting the B-form DNA. This must be a reason of why Top1 plays little or no role in dealing with (+)SC during branch migration [39]. Indeed, Top3 appears suitable for relaxing the double helix during branch migration. A concerted RecQ-Top3 action can enable one strand to pass through the other strand right after the cross-join migrates a few nucleotides. In this way, supercoiling could even be avoided altogether. Thus, in DPE dcHJs, RTR can theoretically bring the two cHJs up to a proximity where no more half helical turns are present (DPE-0) (Figure 4D). Upon reaching this point, cHJs are converted into a series of ssDNA interlinks known as hemicatenanes, which are substrates for Top3 as well. Whether the DPE-0 intermediate exists, and how it is actually processed is unknown; however, converging branch migration of the dcHJ results in dissolution of the JM without any exchange of genetic material [17,18]. It can be speculated that three consecutive Top3 ssDNA decatenating reactions complete DPE-0 dissolution (Figure 4D). The situation is different for the DPO dcHJ (Figure 4E). Here, the smallest distance between both cHJs leaves one helical half-turn (DPO-1), with seemingly strong thermodynamic and topological constraints against it. It is not clear whether this substrate can be further reduced to a DPO-0 by RTR. In case it is, complex nautical-type hemicatenane figures are expected. The complexity depends on the way strands intertwine with each other, which in turn depends on the relative under or over position of cross-over strands in the cross-joins. In Figure 4E, I depict what might be the most complex figure, which, nevertheless, requires just two Top3-mediated strand passage reactions. From a genetic point of view, RTR dissolution of DPE and DPO dcHJs lead to non-crossovers (NCOs)—i.e., no flanking exchange of genetic information between the engaged DNA molecules. In addition, and provided that MMR had not dealt with the initial hxDNA formed between the two cHJs, RTR dissolution would cause no short-tract GCs as it restores the original complementary strand annealing.

By contrast, endonucleolytic resolution of dcHJs that affect two homolog chromosomes do have genetic consequences of the outmost importance. Firstly, the two different planes that resolvases can cut through a single cHJ implies that there are two genetic outcomes when we consider a couple of cHJs (Figure 4F,G). The first outcome yields NCOs. The second outcome results in a terminal exchange of information—i.e., a crossover (CO). Whether NCOs or COs, the distance between cHJs at resolution determines the length of hxDNA. This distance could have changed from the moment of the first and second end capture due to branch migration. It is not known whether there exists temporal and spatial coordination between the resolution of both cHJs. With regard to spatial coordination, deviations from a 1:1 ratio between NCOs and COs would be predicted if present. In addition, this bias ratio would swap when considering DPE versus DPO. For example, if the “b,d” cut orientation were preferred, a DPE would lead to a >1 NCO:CO ratio, whereas <1 ratio would be expected for a DPO (Figure 4F,G). As in the case of hxDNA and GC, the terms NCO and CO make sense when two homolog chromosomes are connected by JMs. An equivalent to CO when two sister chromosomes are involved is the term “sister chromatid exchange”; however, I use here CO/NCO for both homolog and sister chromosomes.

### 2.4. A Double HJ with Two Nicked HJs. The Double Nicked HJ (dnHJ)

Akin to the dcHJ, I consider having a partner of nearby nHJs (dnHJ). In the first scenario, nicks are present in an otherwise model DPE dcHJ (Figure 5A). I can envisage four isomers depending on whether the nicks occur in the same (isomers 1 and 2) or alternate (isomers 3 and 4) crossing strands. As stated above for a single nHJ, only one resolution reaction is possible, leading to NCOs in all four isomers. Furthermore, the nicked products are indistinguishable. Dissolution of dnHJs would also result in NCOs, with a correspondence between the nicked strand(s) in the dsDNA products and the original DPE dnHJ isomer. In the second scenario, one nick would be placed in one strand which does not participate in the cross-joins of a model DPE dHJ. As reasoned above, such nHJ is expected to isomerize so that the homologous strand of the nicked strand is repositioned to make the cross-over, becoming a DPO dnHJ (Figure 5B). Resolution of this molecule would lead to an obligate CO, whereas dissolution would have to deal with the problems I have just discussed for DPO dcHJs. Finally, one favorable aspect of dnHJs dissolution is that nicks left behind could act as swiveling centers for topological relaxation, reducing, yet not eliminating, the need for topoisomerases (Figure 5C). On the contrary, a dnHJ could have a reduced dissolution rate if nHJs act as metastable isomers (thermodynamic sinks).

### 2.5. A Double HJ with Only One Nicked HJs. The Double Canonical-Nicked HJ (dcnHJ)

A final case to consider is the occurrence of one cHJ and one nHJ in the partner (dcnHJ). In such case, the genetic outcome of the nucleolytic resolution (NCO vs. CO) depends on the cut plane of the cHJ (Figure 6A). Dissolution should be possible and occur by pushing the cHJ towards the metastable nHJ. Again, two scenarios are conceivable depending on whether the dcnHJ is DPE or DPO (Figure 6B,C). Regardless, a single Top3 ssDNA decatenation reaction appears to be needed based on the plectonemic nature of the cross annealing of the non-nicked strands. However, I cannot rule out Top3-independent dissolution if such cross annealing occurs and remains as a paranemic junction [51], seemingly improbable in vivo owing to its unfavorable topology and the fact that topoisomerases could turn the junction into a plectoneme [52,53]. In all cases, NCOs without hxDNA tracts are the expected products of dcnHJ dissolution. 

Alternatively, an SSE could cut the nHJ only (Mus81* appears appropriate here), leading to two opposing nicks, which can serve as swiveling centers and energetically favorable sinks for directional branch migration of the cHJ (Figure 6D,E). Furthermore, both nicks should relieve most topological issues associated with branch migration, which, in turn, likely opens the repertoire of helicases capable of performing this greatly simplified dissolution. I cannot foresee major differences between DPE and DPO when this combination of nHJ resolution and ensuing cHJ branch migration eliminates this type of JMs. Again, NCOs without hxDNA are expected.

## 3. Dealing with JMs in DSB Repair

In this chapter, I put the aforementioned JMs into the context of the DSB repair by HR, starting with a brief outline of how the HR pathway works (Figure 7A). I mostly overlook the first steps of HR; good reviews can be found elsewhere [42,54]. DSBs can be part of deleterious DNA insults somatic cells are exposed to or emerge as part of the program to trigger recombination during gametogenesis. Noteworthy, not only can HR deal with DSBs but an alternative repair pathway, the non-homologous end joining (NHEJ), operates in specific contexts (see [55] for a review). To date, HR is present in all living beings and is highly conserved. In this chapter, I refer to HR in the budding yeast *S. cerevisiae* unless stated otherwise. The budding yeast is where HR has been more deeply studied and has certain genetic advantages that have enabled many of the steps and subpathways I describe below to be determined. 

To engage HR for DSB repair, the broken ends must be 5′→3′ resected to yield 3′ ssDNA overhangs (Figure 7A, step 1). At least one of these resected ends actively searches for homology across an unbroken dsDNA molecule, with a clear preference for the sister chromatid in mitosis and the homolog chromosome in meiosis. Many proteins of the Rad52 epistasis group participate at this level, including Rad51, Rad52, and Rad54. Rad51 coats the resected end and makes it competent for the dsDNA invasion, Rad52 facilitates this step, and Rad54 is a helicase/translocase that could propel the search forward or backwards [36,56]. The D-loop is formed when the searching 3′ ssDNA finds and invades the donor unbroken dsDNA (Figure 7A, step 2). If there is heterology between the invading (recipient) and the donor DNA (i.e., homolog chromosomes are engaged), the length of the D-loop determines the initial length of the hxDNA. The D-loop is a reversible structure and several 3′→5′ helicases have been shown the dismantle it [57]. For HR to complete its job, the D-loop must prime DNA synthesis, long enough to overtake the point of breakage in the DSBs. Synthesis brings forth an extended D-loop, which now incorporates the donor sequence at the 3′ end. In theory, DNA synthesis could continue until the end of the chromosome, with either an overextended or migrating D-loop. This HR subpathway is called break-induced replication (BIR), and it might be the only alternative to survive one-ended DSBs, which arise from eroded telomeres or DSBs at anaphase bridges [10]. BIR requires synthesis of the complementary strand through a sort of lagging strand replication behind the migrating D-loop, leading to a half crossover (½CO) with no connecting hxDNA (Figure 7A, BIR). The ½CO would affect the recipient chromosome and would appear as a conservative replication with two newly synthesized complementary strands.

In somatic cells repairing a two-ended DSB, the best way to minimize the exchange of genetic information is to (i) use the sister chromatid as donor and (ii) channel the elimination of the D-loop through the synthesis dependent strand annealing subpathway (SDSA) [58,59]. In the SDSA, the extended D-loop is dismantled, and the ssDNA reanneals with the 3′ ssDNA overhang coming from the other end of the DSB (Figure 7A, step 3 and SDSA). The most remarkable genetic outcome of SDSA is that the donor sequence should remain unchanged, whereas any genetic modification in the recipient DNA is kept in the vicinity of the DSB site as a short-tract hxDNA, within a global NCO product. It is also expected that any GC resulting from MMR over the hxDNA is restricted to the DNA synthesis that occurs during the invasion, since the extended D-loop is short-lived and MMR would deal with the hxDNA generated after SDSA—i.e., the hxDNA formed after the donor sequence incorporated ahead of the invading 3′ end reanneals back with its recipient complementary strand. This implies that eventual GCs should only be found on one side of the DSB site. However, this vision of SDSA is an oversimplification since it is known that the transient hxDNA in the D-loop can be corrected by MMR [60,61].

Even though SDSA appears as a preferred HR subpathway in somatic cells, there are instances where the second broken end anneals with the displaced strand in the donor sequence and primes synthesis as well (Figure 7A, step 4). This event changes the HR subpathway from SDSA to the Szostak’s DSB repair (DSBR) subpathway [58,62]. Although in mitotic cells confronted with unwanted DSBs the SDSA/DSBR ratio is inclined towards SDSA, in meiosis, the repair of many programmed DSBs is channeled through DSBR. If synthesis continues in DSBR until one 3′ end encounters the other resected 5′ end, the resulting JM becomes a nHJ (Figure 7A, step 4, reddish strands). If this happens for both ends, the product becomes a dnHJ (Figure 7A, step 5). Of note, dismantlement of these JMs ought to be possible after the second end capture (Figure 7A, step 4′ and 5′). However, if DNA synthesis took place from the second 3′ ssDNA overhang, the resulting modified SDSA subpathway (2xSDSA) would result in hxDNA on both sides of the DSB (Figure 7A, step 5′). Provided that the DSB was clean (there is no sequence gap between the two ends), the two tracts of hxDNA would be contiguous, changing strands at the DSB site: recipient Watson annealing with donor Crick on one side (dark “hx” letters in the figure), and recipient Crick annealing with donor Watson on the other side (light “hx” letters in the figure).

### 3.1. The Canonical DSBR Model

This model contemplates the immediate ligation of the nick, which turns the nHJ into a cHJ. The canonical DSBR subpathway anticipates the formation of one DPE dcHJ per DSB (Figure 7A; step 6). RTR (STR in budding yeast) dissolution may process this JM directly. Dissolution of dcHJs would minimize genetic exchange, and it is thus a suitable mechanism to complete HR in somatic cells. Indeed, hxDNA is only expected in the recipient molecule (the one that had the DSB). The hxDNA would span the sequences filled in by de novo DNA synthesis on both sides of the DSB, exchanging strands at the break, similar to the 2xSDSA pattern described above (Figure 7A; DIS).

Alternatively, each cHJ can be the target of resolvases. Resolution can yield COs between the DSB flanking sequences. As stated above, the chance for COs ought to be 50% if each cHJ is resolved in the open conformation (Figure 4F). Nevertheless, a closer look at the genetic outcomes exposes subtle differences among the four possible resolution scenarios (Figure 7A; RES). These differences consist of the presence and relative position of hxDNAs and donor sequences (“don” in the figures) coming from the de novo HR-driven DNA synthesis. In the two cutting scenarios that lead to NCOs (“a,c” and “b,d”), the genetic outcomes are similar and there is only a shift in the relative position of genetic exchanges. Thus, NCO products would comprise (i) a donor molecule that now carries a short-tract hxDNA on one side of what it was the DSB site in the recipient molecule and (ii) a repaired recipient molecule that carries donor sequences mirroring the hxDNA in the donor molecule and an hxDNA next to it, the boundary between the donor sequence and the hxDNA being the position of the original DSB. Because the short tract donor sequence is surrounded by the full recipient chromosome in this NCO context, this copy of donor sequences can be considered a direct GC event—i.e., independent of whatever outcome MMR can leave on hxDNA tracts. Regarding the COs products, there is a marked difference in the vicinity of the DSB between the two cutting scenarios (“b,c” versus “a,d”). The critical point lies in whether the counter nicks that resolve the dcHJ occur in the strands with the incoming synthesis (“b,c”) or their templates (“a,d”). In the first case (type 1), just a short-tract one-sided hxDNA spans from the DSB to the CO break point. In the second scenario (type 2), interspaced donor sequences would be found on one side of the DSB site, so that the CO break points for each strand (vertical dashed lines) would demarcate a donor-hxDNA tract around the DSB, both in the recipient and the donor molecule. The interspaced donor in one product would mirror the hxDNA in the other one. For the donor chromosome this would appear as parental DNA fitted between the hxDNA and a clean crossover break point (Figure 7A; lower “a,d” product). It is important to note that these genetic patterns of dHJ resolution are only valid if (i) the DSB is clean and (ii) no branch migration takes place from the D-loop formation to HJ resolution. Both restrictions may rarely apply in vivo; accidental (non-programmed) DSBs would often require 3′ end processing to enable priming, and branch migration can reposition D-loops and cHJs before resolution. When these modifications are taken into consideration, the genetic patterns in the vicinity of the DSB change substantially [63,64].

In conclusion, according to canonical HR models for DSB repair, there is just one way to yield COs, resolution, and three ways to obtain NCOs: SDSA, RTR-based dissolution, and half of the resolution events. Somatic cells repairing unintended DSBs are expected to promote NCO pathways, whereas gametes would favor resolution. Although sizeable genetic consequences are only expected for COs events with either homolog chromosomes or ectopic sequences, NCOs are not neutral in the vicinity of the DSB. From the NCO mechanisms, SDSA and dcHJ resolution are seemingly the least and most risky, respectively.

### 3.2. The DSBR Model with Metastable nHJs

In this alternative DSBR model, nHJs are not ligated to yield cHJs, or, at least, the ligation becomes a kinetic limiting step due to thermodynamic and/or topological requirements. Resolution by Mus81*, along with the other resolvases, is possible over this dnHJ, but only one product is expected, the CO with one-sided hxDNAs (type 1) (Figure 7A,B, RES’). A chief prediction based on the thermodynamics of the nHJs is the DPE to DPO isomerization shown already in Figure 5B. Unless tight topological barriers hamper this isomerization, I predict that a DPO dnHJ, rather than a DPE dnHJ, is the universal intermediate in the DSBR (Figure 7B). If ligation eventually occurs, still a DPO dcHJ would be the intermediate. Dismantlement of the DPO dnHJ through a double SDSA-like mechanism still would yield the same NCO product described for the DPE dnHJ (Figure 7B, 2xSDSA). A metastable dnHJ may, in turn, form a barrier for the dissolution pathway. In case dissolution occurred, it would face the mechanistic issues raised in Figure 4E and Figure 5B,C. Likewise, the uncertain outcome of DPO dissolution would be present if the dnHJ is transformed into a dcHJ (Figure 7B, DIS). Resolution of a DPO dcHJ would yield the same DPE dcHJ products (Figure 7B, RES), with just a shift in the relative relationship between cutting planes and resolution products.

In conclusion, the predictions of metastable dnHJs are: (i) a cross-join isomerization of the nHJ that results from the second end capture; (ii) probable difficulties to channel the elimination of dnHJ and dcHJ through the dissolution pathway; and (iii) therefore, a major prevalence of the resolution pathway, with a prominent role for nHJ resolvase activity (Mus81*, but also Yen1), which would result in a single CO product with a defined genetic pattern around the DSB (type 1). 

## 4. Dealing with JMs during RS

If DSB repair is already ostensibly complex, how HR is involved in RS is intricate as well. I explain two prototypical scenarios that contemplate replication blockage at the lagging and the leading strand, respectively. I acknowledge other RS variations are possible—e.g., interstrand crosslinks and replication fork (RF) collapse after encountering a nick or a gap. However, I skip these for simplification and because the expected JMs either do not differ from those in DSB repair or do not contain nHJs. Likewise, I do not consider replication fork regression, which leads to a transient cHJ-like JM, because its processing leads to either DSBs or JMs that do not comprise nHJs.

### 4.1. Replication Blockage at the Lagging Strand

Replication proceeds in S phase from replication origins in both directions, generating two RFs per activated origin. During certain forms of RS, chemical or proteins adducts can block replication of just one nascent strand. If the lagging strand is the one blocked, the RF could still walk along the chromosome, in theory, with a new lagging strand being synthetized ahead. This results in a ssDNA gap behind the walking RF (Figure 8A, step 1). One way to bypass this blockage is to engage specialized low-fidelity polymerases in the so-called translesion synthesis pathway (Figure 8A, TLS). Alternatively, HR is enlisted to melt the blocked lagging strand from its template and invade the sister chromatid. This template switching forms a D-loop that enables to resume synthesis. From here, the gap can be filled and the extended D-loop dismantled without the displaced strand being captured in a SDSA subpathway as that shown for DSB repair in Figure 7A (steps 3 and SDSA). Noteworthy, the displaced strand in the extended D-loop could anneal with the ssDNA gap to form a DPE dncHJ (Figure 8A, step 2). To accomplish this second capture, it is likely that the displaced strand and the ssDNA gap need to acquire a paranemic conformation, and that a type I topoisomerase acts on the structure to form the plectonemic annealing. Some models place this “second capture” as the first step, such that the ssDNA gap invades the donor; however, the DPE dncHJ is still the predicted product (Figure 8B, consider steps 2 and 3 for a single RF blockage).

Elimination of the DPE dncHJ could be accomplished through four different mechanisms. The first one comprises the combined action of Mus81* to resolve the nHJ, followed by 5′→3′ branch migration of the cHJ towards the two opposing nicks, which serve as a sink to dissolve the cHJ (Figure 8A, steps 3 to 5; see also Figure 6D). A putative alternative scenario would be 3′→5′ dsDNA melting from the nicks by helicases. A second mechanism could rely on 5′→3′ migrating the cHJ towards the nHJ (Figure 8A, steps 6 and ½DIS). The likely requirement for a topoisomerase to enable a plectonemic annealing of the crossover strand that lacks a break (the dark red strand in my example), makes likely that the elimination of the dncHJ by this mechanism needs a type I topoisomerase, supporting the mechanistic view shown in Figure 6B. Because of this, the RTR complex appears suitable for this job, although it should not be considered a proper dissolution pathway but a half dissolution pathway (½DIS). The third mechanism would be a canonical dissolution pathway, which could act before or after the nHJ is ligated (Figure 8A, DIS). From a genetic viewpoint, all these three mechanisms would result in the same NCO products. The fourth and final mechanism involves resolution of the dncHJ, or the dcHJ after nHJ ligation, which would lead to NCOs/CO products on the basis of what I already described in Figure 4F and Figure 6A.

### 4.2. Replication Blockage at the Leading Strand

If the leading strand is the one blocked by the adduct, the RF helicase complex would open the parental strands, but there is no way to re-prime leading strand synthesis ahead. The resulting nascent chromatid would carry a ssDNA tract between the blocked leading strand and the walking RF (Appendix A, step 1). There are at least three ways to bypass the blockage—TLS, fork regression, and template switching. A fourth way is possible provided that a converging fork enables synthesis of the complementary strand, which approaches as a lagging strand for that fork (Figure 8B; dashed oval). In this case, a post-replicative ssDNA gap would emerge, and the same bypass model described in the previous chapter would apply.

If template switching followed by synthesis bypasses the leading strand blockage, the tip of the resulting D-loop could be transformed into a pseudo-HJ by the second capture of the displaced strand (Appendix A, step 2). This pseudo-HJ comprises four arms, but one of them is a ssDNA with a 3′→5′ tail (Appendix A). Interestingly, this JM is also a substrate of Mus81* [65], which cuts at the crossover strand, enabling removal of the remaining cHJ through branch migration with helicases (Appendix A, steps 3 to 5). Despite the presence of the pseudo-HJ, dissolution and half-dissolution by RTR could also be possible in ways not so different to the ones described in the previous chapter (Appendix A, steps 6, 7, RES, DIS and ½DIS; Appendix A). Nevertheless, the main conclusion is that not even a single nHJ is predicted as an intermediary, unless the D-loop reaches a converging RF, which has been described above.

### 4.3. DPE and DPO dHJs in RS

Although D-loops are early commonalities in HR models for RS and DSB repair, there are several differences between JMs that emerge later from these D-loops. In the DSBR subpathway, I predict the DPO dnHJ as the critical intermediate, whilst the DPE dncHJ is the critical one in the RS-driven ssDNA gap repair. Ligation of these JMs into dcHJs would still preserve their basic DPE/DPO configuration. Thus, dissolution and resolution rules apply for each mature JM as described above. Having said that, there are scenarios where RS-driven HR might have to deal with DPO dcHJs. These JMs would emerge from independent RFs which are facing co-temporal RS (Figure 8B). 

It is generally assumed that JMs arising from a single HR event are processed in a concerted manner, independently of JMs that arise from other events in *cis*. Although this might be the case in many instances, a formal proof is missing, and many RFs are expected to get blocked/stalled during RS; each of these RFs could leave an ssDNA gap behind (Figure 8B, step 2). Even blockage of the leading strand could bring about ssDNA gaps when considering there is a converging fork approaching ahead (Figure 8B; dashed oval). Just taking two contiguous RFs, at least six RS scenarios need to be analyzed (four of them are depicted in Figure 8B). In the first one, diverging replication forks encounter DNA adducts located in the same parental strand. This would lead to one leading and one lagging strand blockage, respectively. In the second scenario, diverging RFs would face adducts in opposing parental strands. There are two variants in this scenario, considering that the adducts could block either the leading or the lagging strands from both RFs (Figure 8B only depicts the example for the lagging strand; left RF bubble). The third scenario contemplates two converging RFs encountering adducts on the same parental strand. This would also lead to a leading blockage for one RF and a lagging blockage for its converging partner. The fourth scenario is defined by two converging RFs with adducts on opposing strands. Again, there are two variants depending on whether the leading or the lagging strands are the ones blocked (Figure 8B only depicts the example for the lagging strands; converging RFs on the right). Pairs of ssDNA gaps coming from two adjacent RFs blocked in opposing strands could generate DPO dcHJs by branch migration (Figure 8B, step 3; the grey oval embraces the example of converging RFs).

## 5. Interconverting and Resolving cHJs and nHJs with dsDNA Nicks: The DPO dHJ Gives CO Directionality to Mlh1-Mlh3 Resolution in Meiosis

A particular and intriguing example of CO formation occurs in meiosis through the MMR heterodimer endonuclease Mlh1-Mlh3 (MutLγ). This is essentially the major pathway for CO formation in meiosis; it is highly conserved in evolution, it depends on the synaptonemal complex and the ZMM proteins (Zip1-4, Mer3, and Msh4-Msh5 in *S. cerevisiae*), and the resulting dHJ is resolved only into COs by a mechanism that relies on the Mlh1-Mlh3 endonuclease and the RTR helicase Sgs1 [66]. Intriguingly, Mlh1-Mlh3 does not behave as a typical SSE; rather, it resembles a prokaryotic type IIS nicking endonuclease, with the particularity of recognizing HJ-like JMs instead of a specific sequence [25]. In other words, Mlh1-Mlh3 makes nicks onto dsDNA away from the HJ (around 1 Kb apart). This doubles the complications of this major meiotic pathway: How can this distant nicking activity resolve dHJ? How can this resolution yield exclusively COs? From the mechanisms I have mentioned above, I can formulate several pathways that answer both questions. These pathways rely on nicks as thermodynamic sinks for cHJs/nHJs (Figure 6D,E), and the isomerization of the dnHJ (Figure 7A). Before getting into the pathways, I explore how nicks occurring nearby dHJs affect branch migration.

### 5.1. Effects of dsDNA Nicks in the Vicinity of dcHJs

I consider different scenarios regarding both the number and relative location of nicks. In all examples, I show a DPE dcHJ with inward nicks for simplification. In the next chapter, I also show DPO dHJ and outward nicks for selected cases.

In the first two examples (Figure 9A), only one nick is present between the two cHJs. The nick may locate in one of the crossover strands in the DPE conformation (case 1) or, alternatively, in one of the non-crossover strands (case 2). As shown in other previous JMs, these nicks topologically favor branch migration and may act as thermodynamic sinks. Once a migrating cHJ gets to the nick, it is transformed into a nHJ. In the example 1, the DPE dcHJ should become a DPE dncHJ, whereas in the second example, the nHJ would isomerize to switch the crossover strands into non-crossover strands to become a DPO dncHJ. In either case, the half-dissolution pathway could eliminate the JMs, yielding NCO products with no associated hxDNA tracts.

In the second set of examples, two opposing nicks are considered (Figure 9B). In the first scenario, both opposing nicks occur in the crossover strands (case 1). Branch migration towards the nicks would directly eliminate both cHJs, yielding an NCO with no associated hxDNA tracts. In the second scenario, the opposing nicks locate in the non-crossover strands (case 2). Branch migration, on its own, could again eliminate this dcHJ. Despite strand swaps can be depicted once each cHJ gets to the opposing nicks, the final consequence of the double elimination by branch migration is an NCO without hxDNA tracts. In the third scenario, one nick is present in one of the crossover strands, and the opposing nick affects one of the non-crossover strands (case 3). In such case, the first cHJ that branch migrates towards the opposing nicks generate a one-ended DSB (1e-DSB) for one out of the four dsDNA arms that converge into the cHJ. In this way, the cHJ would become a three-way junction (Y structure), resembling somehow a RF, with the second cHJ still laying ahead. Several JM elimination pathways may deal with this new JMs—e.g., Mus81* could cut the aberrant Y structure. If the second cHJ is eliminated by branch migration towards the Y, a second 1e-DSB would be generated on the same molecule and the same coordinate of the first 1e-DSB, hence effectively becoming a two-ended DSB (2e-DSB). The first 1e-DSB and the 2e-DSB can be further processed by several of the DSB pathways described above (e.g, BIR, NHEJ, SDSA and DSBR). From a physiological point of view, the origin of “case 3” nicks appear uncertain; however, “case 1” and “case 2” nicks could originate by simply resolving other cHJs/nHJs; consequently, there might be good arguments for them in vivo.

In the last and most interesting set of scenarios, four nicks are present along the dcHJ. The combinations of where to place them are increased. I consider seven scenarios (Figure 9C). In the first two cases, both pairs of opposing nicks localize in either the crossover strands (case 1) or the non-crossover strands (case 2). In both cases, cHJs go away into NCO products, with a short tract of interspersed hxDNA delimited by the coordinates of the pair of nicks. The third scenario comprises one pair of opposing nicks in crossover strands and the other pair in the non-crossover strands (case 3). In this case, branch migration resolves the DPE dcHJ into compulsory COs with a tract of hxDNA in the middle. In the fourth scenario, one crossover strand carries two sequential nicks, and their opposing partners reside in one of the non-crossover strands (case 4). Branch migration of one cHJ towards its nearest pair of nicks would break the cHJ into one 1e-DSB and a three-branched Y junction (as in case 3 in Figure 9B). Migration of the second cHJ towards the other pair of nicks would generate another 1e-DSB and another Y-junction, in such a way that one of the parental molecules gets broken twice and the unbroken parental molecule would now carry a double Y junction, which resembles a pair of diverging RFs (a replication bubble). In addition, this double Y would comprise two opposing hxDNA tracts. The two ends of the broken molecule are not continuous in sequence, with a gap that corresponds to the hxDNAs embedded in the unbroken molecule, so that repair by NHEJ, for example, would cause a deletion. In the fifth and sixth scenarios (cases 5 and 6), one pair of nicks occurs in one of the crossover strands and its complementary sister/homolog strand (like in case 4), whereas the second pair occurs in either the crossover strands (case 5) or the non-crossover strands (case 6). The cHJ that migrates towards the first pair of nicks would resolve into 1e-DSB and a Y-junction. The second cHJ is fully eliminated when getting to the second pair, leading to two short tracts of opposing hxDNA, between the Y-junction and the second pair of nicks. The seventh and last scenario comprises one pair of opposing nicks occurring in one of the crossover strands and its complementary sister/homolog strand, with the second pair adopting an equivalent distribution but in the remaining two strands (case 7). In this case, elimination by branch migration leads to two 1e-DSBs, one per parental dsDNA, and a new CO dsDNA branched intermediate. The CO intermediate is complex, formed by two Y-junctions (Y’) with hxDNA in between. Unlike the Y-junctions described for case 4, the strand exchanges of Y’-junctions do not resemble RFs in a replication bubble. Several putative resolution pathways can be envisioned—e.g., nucleolytic resolution, branch migration, helicase-driven unwinding, or combination thereof. Most of these cause the breakage of the CO intermediate, leading to two 2e-DSBs, whose repair can in turn be achieved by NHEJ or HR pathways. 

### 5.2. CO-Biased Models with Mlh1-Mlh3 Making Nicks in Trans

In 2017, Alani’s group reported a critical work on how the meiotic ZMM-based resolution might eliminate dHJs [25]. In this work, they suggested that (i) Mlh1-Mlh3 forms a polymer along dsDNA, starting from a HJ; (ii) once the polymer reaches a critical size, the Mlh1-Mlh3 nicking activity becomes activated; and (iii) in the vicinity of the HJ, the polarity of the parental dsDNAs predicts a counternicking activity in *trans*, so that each parental dsDNA would carry opposing, or nearly opposing, nicks. In this context, Mlh1-Mlh3 would lead to one or several of the scenarios depicted in Figure 9C. Taking into account that catastrophic scenarios (cases 4–7) must be avoided, the counter-nicking activity must be precise and directional, favoring concerted nicking of either crossover or non-crossover strands (cases 1–3). Since obligated COs are the products of Mlh1-Mlh3, cells must inversely regulate the selection of strands that are nicked if dealing with a DPE dHJ—i.e., nicking crossover strands for one HJ and non-crossover strands for the second HJ (case 3). Simpler solutions, such as considering that the Mlh1-Mlh3 polymer has intrinsic specificity for crossover/non-crossover strands (cases 1 and 2), would inevitably lead to NCO products (Appendix A). Nevertheless, this issue can be solved if the target molecule is a DPO dHJ, in which specificity (“a” for crossover strands; “b” for non-crossover strands) reconciles well with compulsory CO products (Figure 10A). As already stated, DPO dHJs (either DPO dcHJ or DPO dnHJ), and not DPE dHJs, are the probable central intermediates of the DSBR subpathway.

In Figure 9, I depict nicks occurring inwards, between the two HJs. In the context of dHJs that emerge following the DSBR subpathway, these nicks would lead to different local genetic outcomes when considering that Mlh1-Mlh3 can nick in either the initial hxDNA tracts formed by strand exchanges or the region filled in by DNA synthesis, which has created donor dsDNA tracts (Figure 10A and Appendix A). Because Mlh1-Mlh3 would nick only once each of the four strands in the DPO dHJ, several “inwards” scenarios are possible. For instance, there could be just one nick upon the newly synthesized tract, with the other three nicks taking place upon the hxDNA tracts in the dHJ (Figure 10A and Appendix A). In addition, this could occur for either one (Figure 10A) or two (Appendix A) of the predefined strand specificities (“a” and “b”). In all these cases, COs with type 1 products would be the outcome. This local genetic arrangement would be equivalent to the one obtained by SSE resolution of dnHJs (Figure 7, RES’). The only differences relate to the rearrangement of the newly synthesized tract nicked by Mlh1-Mlh3, which would now be partitioned between both recombinant molecules, and the presence of a strand swap breakpoint in the hxDNA tract of one of the two solutions (type 1′ CO). Similar type 1 patterns can be achieved by converging branch migration followed by CO-biased SSE resolution (Appendix A, “b,d” and “a,c” cutting planes). Indeed, since disassembly of a HJ by branch migrating towards two opposing nicks restores parental strand annealings (Figure 6D,E), the proposed Mlh1-Mlh3 CO resolution mechanism and the SSE CO-biased resolution upon migrating cHJs can be seen as conceptually equivalent in terms of genetic products. When inward nicking occurs in hxDNA tracts for the four strands, a new local outcome is expected (type 3 in Appendix A). This outcome comprises asymmetric hxDNA-donor-hxDNA and can only occur for one out of the two specificities (“a” or “b”), and when Mlh1-Mlh3 nicks dsDNA either too close (Appendix A) or too far (Appendix A) from its nucleating HJ. The other alternative outcome is still type 1 COs. Two rather specific inward nicking scenarios deserve a final consideration. The first one is a particular case of the one shown in Appendix A, and contemplates that the Mlh1-Mlh3 polymer reaches and nicks upon the other HJ (Appendix A). For a DPO dcHJ, HJ resolution must take place with opposing strand crossover preference for nucleation and incisions, respectively, to yield type 1 COs (Appendix A, “b” specificity). Appealingly, a DPO dnHJ does not need such restriction and just needs Mlh1-Mlh3 to nick upon the single crossover strand (Appendix A). The second particular scenario considers that one of the Mlh1-Mlh3 polymers nicks at what was the DSB site, so that it incises at the junction between the initial hxDNA formed by the invasion and the newly synthesized tract (Figure 10A, Mlh* complex on the right). In this case, type 1 CO molecules are the outcome despite neither newly synthesized tract is nicked, being the only exception to the first rule I describe in this paragraph.

Inward nicking can, however, be too risky in Mlh1-Mlh3 resolution as they can generate DSBs if the two nicks occurring in the strands of the same molecule are very close one another. Safer patterns of Mlh1-Mlh3 can be envisaged if the polymers face outwards or, otherwise, align in just one direction (Appendix A, respectively). Outward cutting would lead to a single local genetic pattern, regardless of the crossover/non-crossover preference, and with a certain degree of symmetry for both CO products. In all cases, the region that delimits the CO would comprise an hxDNA-donor-hxDNA tract (Appendix A). When both recombinant products are compared to each other, a symmetric pattern is expected whereby the sum of one hxDNA-donor half in one CO molecule mirrors the remaining hxDNA half in the other product (type 4 CO). The third scenario, the one-direction resolution, would leave a local outcome that is a mixture of the inwards and outwards models (Appendix A), with the inward cutting having the variability described above. These can end up in recombinant molecules having complex and asymmetric local genetic patterns (type 5 and 6 in the example). As with the inwards scenario, these two alternative scenarios are conceptually equivalent to divergent and co-directional branch migration preceding CO-biased SSE resolution, respectively (Appendix A).

After the first draft of this manuscript was finished, two back-to-back papers in Nature led by Hunter and Cejka groups provided additional insights into the Mlh1-Mlh3 mechanism, as well as how the nuclease could pick the correct strands for CO-biased resolution [23,24]. Mlh1-Mlh3 does not work alone but as part of a multimeric complex that resembles the canonical MutLα MMR complex, and includes Msh4-Msh5 (MutSγ), Exo1, the proliferating cell nuclear antigen (PCNA), and the replication factor C (RFC). PCNA is a replicative clamp that accompanies DNA polymerase δ (Pol δ) during DNA synthesis, and RFC loads PCNA onto DNA. Now, the authors show that RFC/PCNA interacts with Mlh1-Mlh3 and is critical for both Mlh1-Mlh3 activity and CO-biased resolution. Since, RFC/PCNA is asymmetrically distributed towards the newly synthesized tracts, this may provide a clever mechanism to direct Mlh1-Mlh3 incisions into crossover strands (“a” specificity) in DPO dHJs, as depicted in the inward nicking models of Figure 10A and Appendix A. In all cases, type 1 COs are the genetic outcomes. Strikingly, by targeting Mlh1-Mlh3 incisions to the newly synthesized tract, the restriction of crossover/non-crossover strand specificity can be bypassed. Thus, PCNA could orchestrate Mlh1-Mlh3 to yield type 1 COs for both DPO and DPE dHJs (Figure 10B and Appendix A). The same principles I have just described for the different local genetic outcomes of COs apply for DPE dHJs in this case.

### 5.3. CO-Biased Models with Mlh1-Mlh3 Making Nicks in Cis

Hunter’s paper presents an alternative model to those I have just introduced for Mlh1-Mlh3 resolution [24]. In this model, Mlh1-Mlh3 incisions occur in *cis* rather than in *trans* (Figure 11A, left branch). Elegantly, their model evokes the 3′-nick directed MMR subpathway, whereby MutLα makes a single-strand incision 5′ to the mismatch while using a pre-existing nick located 3′ to the mismatch as a reference point (Appendix A) [67]. This pre-existing nick is recognized by RFC/PCNA, which then activates MutLα to make the incision so that the mismatch is flanked by nicks on the same strand. Next, the 5′→3′ exonuclease Exo1 degrades from the 5′-nick to the 3′-nick, enabling Pol δ to re-synthetize the tract and correct the mismatch. Alternatively, Pol δ can synthetize from the 5′-nick and displace the nicked strand in an Exo1-independent manner. From past and recent data, the nuclease activity of Exo1 appears dispensable for stimulating incisions by Mlh1-Mlh3 in vitro and COs in vivo [23,24,68]. This might leave re-synthesis plus strand displacement by Pol δ as the most likely activity to complete cHJ resolution in vivo (Appendix A). Alternative scenarios based on helicases unwinding the HJ from the incisions followed by Pol δ filling in the resulting ssDNA gap cannot be ruled out at present (Appendix A). Because of the crossover nature of the HJ, both a 3′→5′ helicase (e.g., RecQ-like) and a 5′→3′ helicase would be required. In either case, 5′- and 3′-flaps are expected as intermediates and flap endonucleases need to be introduced as players in this pathway. Simpler scenarios that revolve around branch migration cannot directly resolve cHJ with incisions in *cis*; rather, transitions between cHJs and two adjacent nHJ solutions are expected (Appendix A). However, the presence of a nick adjacent to the nHJ may provide a substrate for helicases to resolve the nHJ while generating a flap (Appendix A, top branch for an example with a 3′→5′ helicase and/or Pol δ). When a single HJ is processed by incisions in *cis*, the second HJ in a dHJ can be transformed into a nHJ, a pseudoHJ, or resolved through half-dissolution (Appendix A). HJ disassembly through migration to opposing nicks is also possible (Appendix A, top branch).

To resolve dHJs in a 3′-nick directed MMR-manner, each HJ needs to be repositioned so that incisions in *cis* flank the HJ. This could be achieved by converging branch migration by RTR (Figure 11, conBM step) [24]. In their model, a dcHJ is the substrate; hence, a double incision by Mlh1-Mlh3 is needed, with RFC/PCNA remaining bound to the newly synthesized end after ligation (Figure 11A, left). However, this vision deviates from the 3′-nick directed MMR subpathway. Moreover, double incisions on a dcHJ ought to yield non-type 1 COs. For instance, if the incision near PCNA locates into the newly synthesized tract, a one-sided specular (from the DSB site) hx-recipient-hx CO pattern (type 7; Figure 11A, left) is expected. In order to achieve a type 1 CO, the incisions should occur out of the newly synthesized tract (Appendix A)—a scenario not predicted for the expected PCNA-MutL polarity. Alternatively, Pol δ could nick translate beyond the 3′n, eliminating the outer hxDNA-recipient tract (Appendix A). 

A closer analogy to the 3′-nick directed MMR subpathway can be obtained with a dnHJ, in which nicks are maintained after branch migration, as represented in Figure 5C, and provide a binding/recruiting point for RFC/PCNA. In this scenario, a single upstream incision is required, and a type 1 CO is the final outcome (Figure 11A, right). 

As suggested in previous chapters, nicks close to cHJ likely act as thermodynamic sinks. Thus, branch migration could move the cHJ back to the pre-existing nick at 3′ just after the Mlh1-Mlh3 incision and prior to downstream MMR-like steps. In addition, the cHJ could branch migrate towards the new nick at 5′. In both cases, the cHJ turns into a nHJ, and the remaining nick can seed unwinding by helicases or, else, strand displacement during nick translation by Pol δ (Appendix A, top and bottom). Variations of these motives can be envisaged from a DPO dnHJ by converging, diverging, and co-directional branch migration (Figure 11B). A type 1 CO is the only solution for all cases; the differences lie in the partition of the newly synthesized tracts and position of resulting nicks. By similarity with the 3′-nick MMR system, divergent branch migration appears as the most appealing scenario, with the new nicks 5′ to the nHJs serving as starting points for either Pol δ or 3′→5′ helicases. Other branch migrations would require Pol δ to start nick translation from at least one nHJ, or the unwinding by any of the less common 5′→3′ helicases.

In all models of Mlh1-Mlh3 resolution in *cis*, a DPO dHJ needs to be invoked. DPE dHJ would lead to double incisions in a non-crossover strand for one out of the two cHJs (Appendix A). This HJ could not be resolved by any of the aforementioned strategies, resulting in non-disjunction of the homolog chromosomes unless an SSE intervenes. In the unlikely case that resolution is achieved by shifting the polarity to the template strand for the newly synthesized tract, NCO would still be the expected outcome (Appendix A). Finally, and as already stated, Mlh1-Mlh3 could resolve dHJ by making incisions in *cis* around just a single cHJ (Appendix A, top branch). However, this scenario would still render NCOs for both DPE and DPO dHJs (Appendix A).

## 6. Evidence for and against Metastable Nicks and nHJs

Probably the main argument against metastable nHJs in the DSBR and RS models comes from the historic viewpoint of these models. Many previous studies have dealt with the different HR steps shown in Figure 7 and Figure 8, but almost none with the “ligation step”. This step is often taken for granted. One reason for this is that nicks are considered too risky for genome stability. Another reason is that cHJs in the dHJ models effectively explained the early NCO:CO ratios observed in meiosis [58]. However, the DSBR model with cHJs, and the corresponding cutting planes that yield NCOs/COs, was formulated before other NCO pathways (SDSA and dissolution) and CO pathways (meiotic ZMM) were introduced. 

As described in previous chapters, from a conceptual point of view, ligation of the nick in the nHJ is not as simple as it appears. The nicked strand in the nHJ cross-join is probably subjected to both thermodynamic and topological constraints, which need to be overcome for sealing the nick. Ligase may operate well on dsDNA nicks but not necessarily on the nHJ. To the best of my knowledge, this has not been addressed in model nHJs and dnHJs in vitro, especially in the stressed parallel conformations. On the other hand, genetic studies in *S. cerevisiae* have shown that ligase mutants are fully competent for mating-type switching—a HR-driven GC model [69]. However, since this GC relies entirely on SDSA [70], an extrapolation for the DSBR subpathway is still required. Incidentally, DNA ligase I is dispensable for resistance to irradiation by γ-rays—a known source of DSBs whose repair is often channeled towards the DSBR subpathway [71]. If ligase cannot ligate nHJs, an alternative transformation into cHJs could be achieved by branch migration. In vitro, such branch migration is possible, though with thermodynamic restrictions [49]. Whether this also happens in vivo, and more specifically in stressed conformations such as DPE/DPO dHJs, is more ambiguous. RTR must be able to branch migrate from nHJ in order to fulfil dissolution of the proposed dnHJ intermediate, and will probably require large conformational and topological changes towards open/antiparallel conformations in the resulting dcHJs isomers.

Without RTR intervention, branch migration could be highly impaired and the dnHJ could be relatively stable at the coordinates where it originates. Two predictions can be drawn from this. Firstly, the DPE to DPO cross-join isomerization amply described above, and secondly, a resolution bias towards COs with just one out of the two possible local genetic outcomes (Figure 7B, RES’ type 1 pattern). Outstandingly, resolution towards biased COs with this specific pattern is found in several seminal studies in yeast [63,64,72,73,74,75,76]. This class of genetic studies should be considered carefully, though, as many other results defy the classical view of DSB repair pathways (even SDSA), and deep complexities underlie HR-driven DSB repair (e.g., D-loop and HJ migration, 3′ end trimming, D-loop multi-invasions, template switching, nick translation, hxDNA MMR correction on intermediates, as well as resolved products) [60,61,63,64,77,78]. A description of such intricacies is beyond the scope of this work but, in any case, most studies in different organisms conclude that the DSBR subpathway favors COs with a bias towards the local pattern expected for DPO dnHJ resolution (type 1 CO), whereas NCOs are the result of SDSA and, to a lesser extent, the dissolution pathway [73,74,75,79,80,81,82]. All these strengthen the arguments given in this work. 

These genetic studies include investigations on COs resulting from DSB repair in both mitosis and meiosis. Nevertheless, the meiotic scenario deserves a separate consideration because of the ZMM CO subpathway. Thus, previous studies that used meiosis as a model to refine the DSBR subpathway may need to be revisited as DSBR appears subsidiary to ZMM. In this regard, type 1 CO would often be predicted when Mlh1-Mlh3 nick the four strands between the two HJs (either cHJs or nHJs) in the DPO dHJ (Figure 10, Appendix A). What is needed for type 1 CO is inward nicking of the dHJ with strand specificity, as long as at least one nick occurs in the newly synthesized tract. The strand specificity of Mlh1-Mlh3 can be given by nicking with either crossover or non-crossover preference in DPO dHJs. Nicking upon the newly synthesized tracts during the formation of the dHJ also predicts type 1 COs for both DPO and DPE dHJs. In addition, these models, with Mlh1-Mlh3 making incisions in *trans* upon homologous strands, combine metastable opposing nicks with branch migration of HJs and bring together one of the novel mechanistic views of Mlh1-Mlh3 resolution [25], the preferent type 1 CO outcomes, and the ostensible branch migration that precedes resolution [64]. Very recent models with Mlh1-Mlh3 incisions in *cis* also predict type 1 COs provided that a few restrictions apply (Figure 11 and Appendix A) [23,24]. Remarkably, the DPO dnHJ results in mandatory type 1 COs while following a mechanistically credible resolution strategy founded on the 3′-nick directed MMR subpathway. Interestingly, non-type 1 COs in these models in *cis* could comprise transfer of genetic information from the recipient to the donor molecule (Figure 11A and Appendix A), an intriguing event not predicted in classical DSBR pathways yet observed in several genetic studies, where it has been reported as an artifact ([64] and references therein). The use of an MMR-based mechanism for processing dHJs is a remarkable example of evolution towards multitasking, but it might not be the only one. *Drosophila melanogaster* heavily relies on the MEI-9-ERCC1 endonuclease for COs in meiosis [83]. This endonuclease participates in the nucleotide excision repair pathway, which mechanistically shares many analogies with MMR.

The DPO dnHJ also fulfills the expectations for the Mus81* resolvase—i.e., a single ssDNA cut upon the single crossover strand in each nHJ. The overwhelming data on the substrate specificity of Mus81* for JMs other than cHJs [20,22], and the genetic data that place this resolvase on top of the others [45,46,47], led to a previous revival of the dnHJ as a real and metastable intermediate in the DSBR subpathway [84], although there have been attempts to reconcile the poor activity of Mus81* upon cHJs with dcHJ models. In particular, in higher eukaryotes, Mus81* can be found within a macromolecular complex with other SSEs such as SLX4-SLX1, so that SLX4-SLX1 could make the first nick, transforming the cHJ into a nHJ for the final Mus81* resolution [85]. In addition to this, *S. pombe* Mus81-Eme1 can be posttranslationally modified to enhance its cutting efficiency towards cHJs [86], a circumstance that may be explained by the fact that this yeast lacks a Yen1/GEN1 homolog.

Contrary to the evidence for DPO dnHJs, biochemical experiments aiming to determine the physical nature of JMs arising from both meiotic and mitotic DSBs support a DPE dcHJ as the central JM in DSBR, especially in meiosis [87,88]. In meiosis, these DPE dcHJs probably correspond to those that are resolved by the ZMM CO subpathway, although a distinction from those revolved by SSEs was not recognized at that time. If the DPE dcHJ is the Mlh1-Mlh3 substrate, either the enzyme needs to concertedly change crossover/non-crossover strand specificity when processing each cHJ in the dHJ, or the ZMM subpathway would promote a DPE to DPO isomerization at a stage close to the resolution. For the former, targeting Mlh1-Mlh3 to the newly synthesized tracts by PCNA [23,24], followed by making incisions *in trans* [25], appears to be the only strategy (Figure 10B). In mitosis, the dcHJ was determined in wild type cells for the major SSEs. Thus, it is conceivable that they only detected a subpopulation of JMs unable to be processed by Mus81* and other nHJ resolvases. In this respect, we reported the presence of mitotic nHJs in *mus81 yen1* double mutants [6]. However, it could be argued that our work dealt with RS JMs, rather than those generated after DSBs, and that we looked at the ribosomal DNA array, which owns many unconventional features [10,89]. Further studies are needed to shed more light on these issues.

## 7. Conclusions and Perspectives

Here, I have focused on nicked strands as part of the HR substrates that need to be removed in order to segregate chromosomes in mitosis and meiosis. Nicks can affect one or several of the strands involved in the formation of JMs and can be found within the JM (nHJ) or its vicinity. These nicks have profound implications on the biophysical, biochemical, and topological properties of JMs, as well as determining distinct mechanistic views and genetic outcomes during DSB repair and RS bypass. 

The final steps of HR are still far from being fully understood. I have presented several schematics on how JMs can be processed, comparing canonical models with uninterrupted strands with those that include nicks and nHJs. In these schematics, I have labelled each of the four strands with a different color, with distinct marks for newly synthesized tracts, hxDNA strand composition, strand orientation, and presence of the 3′ end in nicked strands. In all cases, the final recombinant products are presented before the nicks are sealed and the hxDNA resolved by MMR. In this manner, investigators can use the schematics for their own predictions on CO/NCO local maps: (i) CO breakpoints; (ii) direct GC tracts; (iii) presence, position, and strand composition of hxDNA tracts; (iv) presence, position, and partition of the newly synthesized tracts; and (v) presence and position of nicks as a result of JM resolution or other JM elimination pathways. Many of the chief predictions introduced in these schematics have not been addressed yet—e.g., strand composition in hxDNAs, partition of the newly synthesized tracts, and nicks in the immediate HR products, as well as their JM precursors. It is extremely challenging to characterize these intermediates, but it would be extremely informative to confirm or reject the models I have introduced here. Likewise, a better understanding of the molecular mechanisms that processed JMs (SSE and Mlh1-Mlh3 resolution, RTR dissolution, and D-loop disassembly) should refine the presented models.

## Figures and Tables

**Figure 1 genes-11-01498-f001:**
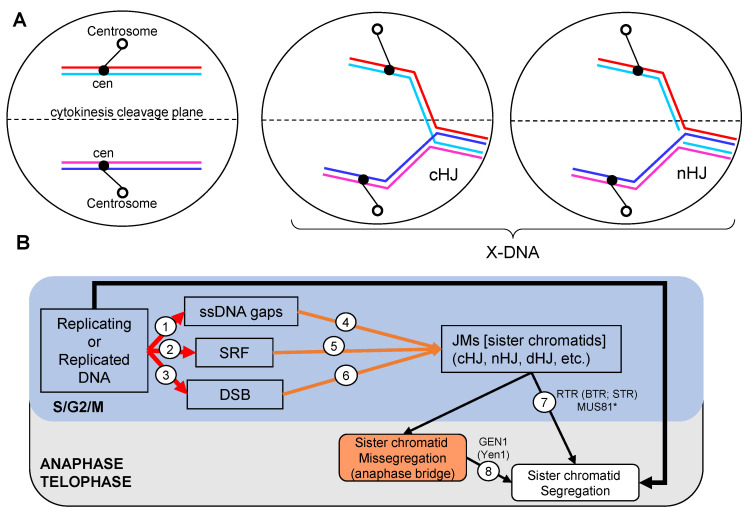
Joint molecules (JM) interfere with chromosome segregation. (**A**) Examples of aborted chromosome segregation due to the presence of a single JM on one chromosome arm. I depict the two major X-shaped JMs considered in this work: the canonical Holliday junction (cHJ) and the nicked Holliday junction (nHJ). The lack of resolution of the distal part of the chromosome arm results in anaphase bridges that can be broken during cytokinesis. On the left, faithful segregation is accomplished when no JMs are present. (**B**) The origin and resolution of JMs in the context of the mitotic cell cycle. Replication stress is common even in an unperturbed cell cycle, leading to single-stranded DNA (ssDNA) gaps (1), stalled replication forks (SRFs) (2), and DNA double strand breaks (DSBs) (3). DSBs may also occur pre- and post-replicatively through several mechanisms. At the time of replication (S-phase) or right afterwards (G2/M), cells can deal with ssDNA gaps, SRFs, and DSBs through specialized DNA repair pathways that depend on the homologous recombination (HR) machinery (4,5,6), which create transient links between two DNA molecules—i.e., JMs. JMs are, in turn, eliminated through several mechanisms that are hierarchically regulated. These include dissolution of double cHJ by the RecQ-like Top3 complex (RTR) in S/G2 (7), and resolution by structure-selective endonucleases (SSE), such as Mus81* in G2/M (7) and GEN1/Yen1 in anaphase (8).

**Figure 2 genes-11-01498-f002:**
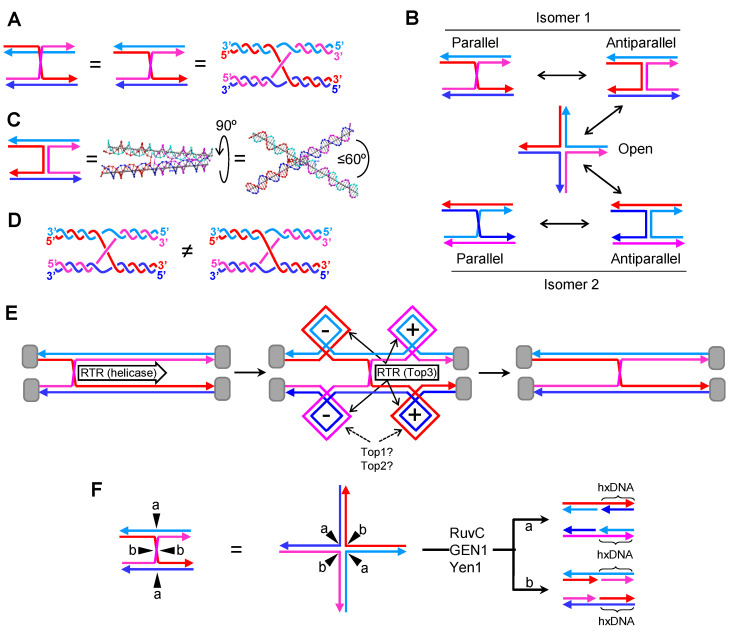
Features of the canonical Holliday junction (cHJ). (**A**) Classical 2D schematics of a cHJ in the parallel conformation. Left: Watson and Crick strands are depicted as straight lines, with the Watson strand on the top in each molecule. Center: the strands of the upper molecule are vertically flipped so the cross-join is easier to visualize. Right: a plectonemic drawing of the strands whereby the double helix of the DNA can be inferred more easily. (**B**) The cHJ can exist as two isomers, which depend on the pair of strands that forms the cross-join. Each isomer can adopt two extreme conformations, parallel and antiparallel. An open planar conformation serves as an intermediate between isomers and conformers. (**C**) The antiparallel conformation with tilted arms in the most thermodynamically favored stereoisomer. The antiparallel conformer is depicted as straight lines (left), or modelled with Nanoengineer-1 on the same XY plane (center) and after turning 90° on the Z-plane (right). (**D**) The spatial position of the strands in the cross-join results in different topoisomers. On the left, the Watson strand of the upper molecule crosses under the Watson strand of the lower molecule. On the right, the crossing plane is inverted. (**E**) Branch migration in topologically constrained molecule ends (grey boxes) results in positive (+) supercoiling ahead and negative (−) supercoiling behind. The putative activities of the RTR complex in each step are indicated (helicase or topoisomerase). Top3 could deal with supercoiling alone or with the aid of Top1 and/or Top2. (**F**) The Holliday junction is removed by resolvases by a nicking and counter-nicking mechanism. Two cutting planes are feasible (“a” and “b”). In the open conformation, each plane should have the same chance, so that two distinct products are equally possible. Resolution of a single cHJ gives rise to two new molecules with one flank identical to the parental DNAs and the other flank with cross annealing of Watson and Crick strands coming from different parental DNAs—i.e., heteroduplex DNA (hxDNA). Curly brackets mark out the hxDNA region in each product. Blueish lines, Crick strands; reddish lines, Watson strands. Dark (blue, red) and light (cyan, pink) colors are also included so that the four strands can be differentiated. Parental dsDNA anneals a light and a dark strand, either cyan with red or pink with blue; whereas hxDNA anneals either light–light (cyan–pink) or dark–dark (red–blue) strands. The arrowhead indicates the 3′ end. The equal sign links different visual schematics of the same molecule.

**Figure 3 genes-11-01498-f003:**
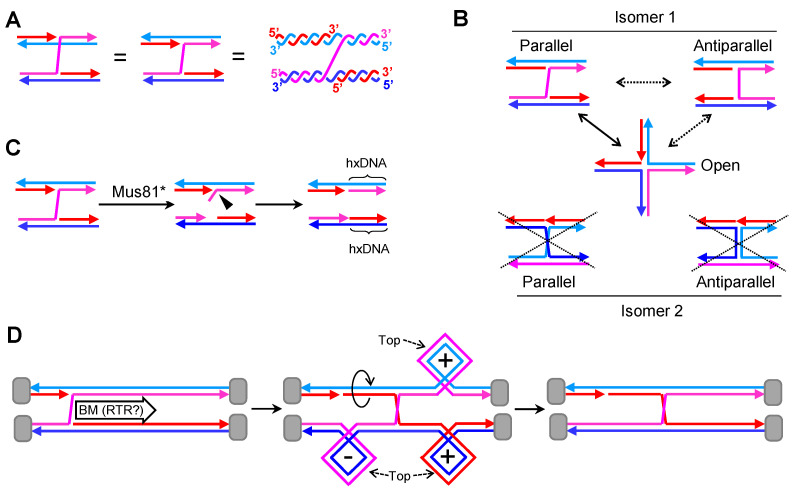
Features of the nicked Holliday junction (nHJ). (**A**) 2D schematics of the nHJ in the parallel conformation. (**B**) The nHJ can exist as one isomer, in which the crossover strand corresponds to the homolog to that with the nick. The nHJ tends to adopt the open conformation. (**C**) The nHJ is the target of a specialized nHJ resolvase, Mus81*, which cuts the crossover strand near the cross-join, leading to two daughter molecules, one with a 5′ flap and one with a gap. The flap and the gap can be further processed in vivo, so the resulting molecules are like one of the two solutions shown in Figure 2F, including the hxDNA flank. (**D**) Branch migration can isomerize a nHJ into a cHJ. In topologically constrained molecule ends (grey boxes), branch migration results in positive supercoiling (+) ahead and negative supercoiling (−) behind the migration direction. In principle, the nick left behind in one strand may relieve negative supercoiling through swirling (turning arrow). For other details see the legend of Figure 2. BM, Branch migration.

**Figure 4 genes-11-01498-f004:**
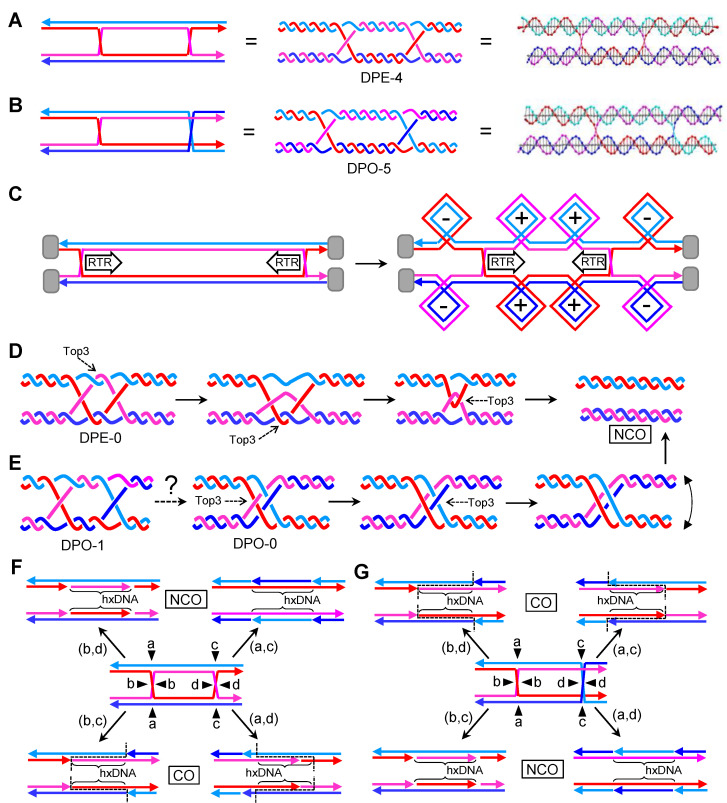
Features of the double canonical Holliday junction (dcHJ). (**A**) Classical 2D schematics of the dcHJ depicted in DSBR models. The double cross-joins involve the same two strands (Watson strands in this example). This kind of dcHJ spans an even number of half helical turns. The schematics depict the parallel conformation with 4 helical half turns between the cHJs (DPE-4; D stands for dHJ, P for parallel conformation and E for even number of helical half turns). (**B**) 2D schematics of an dcHJ isomer where each cHJ has a different cross-join partner (Watson strands in the first cHJ and Crick stands in the second cHJ). This isomer changes the number of helical half-turns to an odd number; in this example, there are 5 half-turns (DPO-5; O stands for odd). (**C**) Converging branch migration of dcHJs results in twice more positive (+) and negative (−) supercoiling than migrating a single cHJ the same distance. (**D**) Converging Branch migration of a DPE dcHJ to zero half-turns gives rise to hemicatenanes that tie the four strands as in a sailing knot. The complexity of the knot depends on the cross-join topology (Figure 2D). Top3 can unlink the strands as in the example so that the dcHJ is dissolved. (**E**) Converging Branch migration of a DPO dcHJ to zero would give rise to a topologically distinct sailor knot. To reach a DPO-0, branch migration should take place in steps of half-turns or less, so that the DPO-1 to DPO-0 transition is feasible. (**F**) The resolution of the DPE dcHJ and its genetic outcomes. Depending on the cutting planes, the resolution can end up in products with or without a terminal genetic exchange—i.e., crossover (CO) or non-crossover (NCO), respectively. Regardless of the product, resolution results in hxDNA between the cHJs (regions with brackets, and also dash lines in COs). (**G**) The resolution and genetic outcomes of the DPO dcHJ are similar than the DPE dcHJ, albeit with swapped NCO/CO products for the indicated cutting planes. For other details see previous figure legends.

**Figure 5 genes-11-01498-f005:**
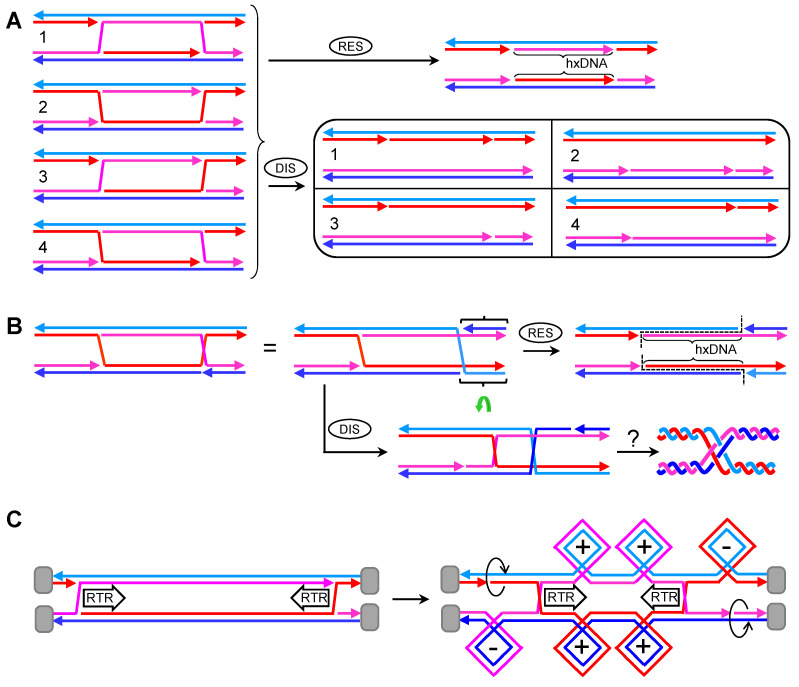
Features of the double nicked Holliday junction (dnHJ). (**A**) 2D straight line schematics of four DPE dnHJ isomers comprising different combinations of nicked strands. In isomer 1 and 2, one out of the four strands carries two nicks. In isomers 3 and 4, Watson strands carry one nick each. All isomers lead to a single resolution (RES) NCO product with hxDNA between the resolved nHJs. In addition, a single pattern of four nicks is expected. Dissolution (DIS) of all isomers lead to NCOs without hxDNA. The relative position of the two nicks is distinct and may serve as a footprint to infer the original dnHJ. (**B**) Two-dimensional schematics of a DPO dnHJ isomer. Since the nHJ isomer with the nick in one of the non-crossover strands is highly unfavorable, the more accurate representation is the one shown in the center, with the right edge vertically flipped to represent the single strand in the cross-join (brackets with the green arched arrow). Resolution leads to a single CO product with hxDNA between the resolved nHJs. Dissolution suffers from the problems stated in Figure 4E. (**C**) Converging Branch migration of a dnHJ would lead to dcHJ with the same positive supercoiling issues raised in Figure 4C. However, negative supercoiling could be relieved as a result of the strand swirling at the nicks left behind. For other details see previous figure legends.

**Figure 6 genes-11-01498-f006:**
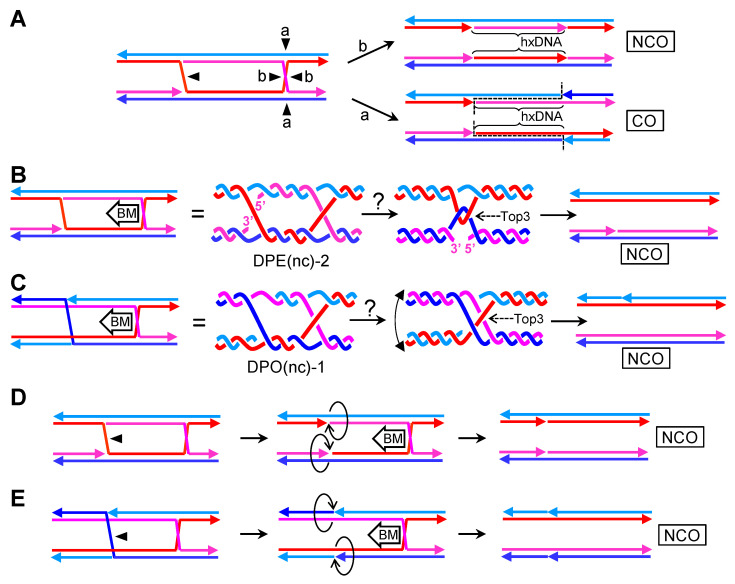
Features of a double Holliday junction formed by a canonical and a nicked Holliday junction (dcnHJ). (**A**) Resolution of a DPE dcnHJ isomer. The NCO or CO product depends on the cutting plane of the cHJ (“a” or “b”). In both cases a short-tract of hxDNA spans between the resolved HJs. (**B**) Dissolution of a DPE dcnHJ by branch migration of the cHJ towards the nHJ. A final Top3-mediated ssDNA decatenation reaction appears necessary. An NCO product without hxDNA and with one single nicked strand is expected. (**C**) Dissolution of a DPO dcnHJ by branch migration of the cHJ towards the nHJ. Top3-mediated ssDNA decatenation could also be necessary. A similar NCO product is expected. (**D**,**E**) The dcnHJ can be eliminated through nHJ resolution (black arrowhead) followed by branch migration of the cHJ towards the resulting two opposing nicks. Branch migration might be favored by the absence of positive supercoiling ahead because nicks would enable strands swirling (arched arrows). The NCO product carries two opposing nicks, footprinting the original strands involved in the cross-join. A DPE dcnHJ is shown in (**D**), and a DPO dcnHJ in (**E**).

**Figure 7 genes-11-01498-f007:**
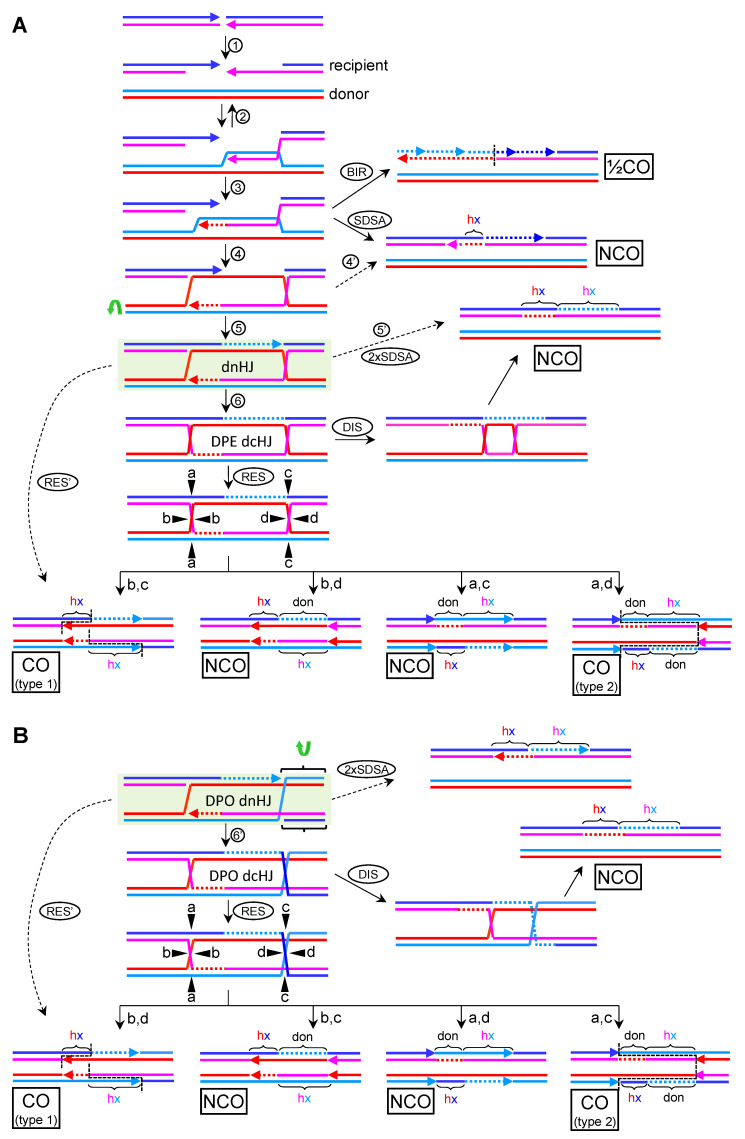
HR pathways for DSB repair. (**A**) Canonical model with DSBR through DPE dHJs. Steps: (1) Resection of DSB 5′ ends; (2) one 3′ overhang invades a donor homologous sequence to form a reversible D-loop; (3) the D-loop primes de novo DNA synthesis (red dotted line); (4) the second DSB end is captured by the displaced strand; (5) the second capture also primes de novo DNA synthesis (cyan dotted line), and when synthesis from both ends cover the resected tracts, a dnHJ is formed; (6) ligation of the nicks results in a DPE dcHJ; (DIS) dissolution of the dcHJ by RTR results in NCOs with two-sided heteroduplex DNA in the recipient molecule; (RES) resolution by SSEs can result in four different products, two NCOs and two COs, with different local genetic patterns (see main text for further details). Deviations from the DSBR model: (BIR) DNA synthesis from the D-loop continues until the end of the chromosome, lagging-type synthesis makes the complementary strand; (SDSA and 4′) the D-loop is dismantled after some DNA synthesis, the extended strand reanneals with its parental complementary strand and further DNA synthesis fills in the ssDNA gaps; (2xSDSA/5′) the dnHJ is disassembled and strands reanneal with their parental counterparts; (RES’) SSEs cut the dnHJ resulting in just a single resolution product (type 1 CO). (**B**) The alternative DSBR model with DPO dHJs. This model includes the cross-join isomerization expected for the second nHJ, resulting in a DPO dnHJ; (6′) ligation would yield a DPO dcHJ. From a genetic point of view, the products of SSE resolution (RES and RES’), RTR dissolution (DIS), and dnHJ disassembly (2xSDSA) are identical to those of DPE dHJs. Other subpathway abbreviations: BIR, break-induced replication; SDSA, synthesis dependent strand annealing. Abbreviations for global genetic products: NCO, non-crossovers; CO, crossovers; ½CO, half crossovers. Abbreviations for local genetic rearrangements: hx, heteroduplex DNA tract (delimited by brackets; the color of each letter encodes the strands involved in the heteroduplex); don, donor sequence tract. The green arched arrow indicates a visual flipping of DNA molecules, or part of them (brackets indicate the flipping portion). The black arrowheads and attached lower case letters point to the nicking planes during resolution. For other abbreviations and signs see previous figure legends.

**Figure 8 genes-11-01498-f008:**
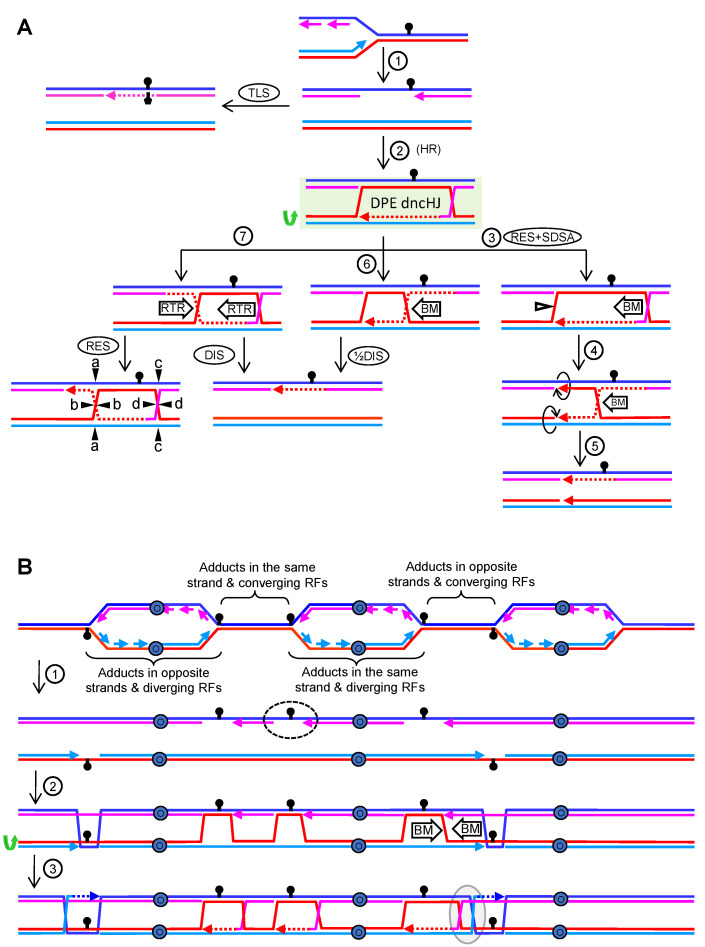
Bypass of replication blockage. (**A**) Gap repair during RS: (1), a DNA adduct (black bulb) blocks synthesis of the lagging strand during replication, leaving a ssDNA gap behind the RF; (2) the 3′ end melts and invades the sister chromatid, which acts as a donor to prime DNA synthesis and bypass the blockage, while the displaced strand is captured by the ssDNA gap, resulting in a DPE dncHJ; (3–5) the JM is processed by first resolving the nHJ with an SSE, followed by branch migration of the cHJ towards the opposing nicks (RES+SDSA-like); (6) alternatively, the cHJ could migrate towards the nHJ and both are eliminated through a half dissolution pathway (½DIS); (7) the nHJ is ligated into a cHJ, and the resulting DPE dcHJ is processed as in Figure 7. The alternative non-HR translesion synthesis pathway (TLS) is also depicted. (**B**) DPO dcHJ can emerge through branch migration from two contiguous DPE dHJs. DNA adducts are expected to spread along a replicating DNA molecule. Several combinations regarding the relative position of adducts, direction of RFs and leading/lagging strand blockage are possible (indicated with brackets). (1) Replication termination would leave ssDNA gaps, even for leading strand blockage (dotted circle); (2) gap repair would proceed as in panel A to yield tandem dncHJs, which can be orientated as nHJ-cHJ-nHJ-cHJ (head-to-tail), cHJ-nHJ-nHJ-cHJ (head-to-head) or nHJ-cHJ-cHJ-nHJ (tail-to-tail); (3) Diverging branch migration can put in proximity two cHJ coming from different bypass events, resulting in DPO dcHJs when two adjacent dncHJs where in either head-to-head or tail-to-tail orientation (grey oval for an example coming from a tail-to-tail partner). An “O” within a dark blue circle points to replication origins. For other abbreviations and signs see previous figure legends.

**Figure 9 genes-11-01498-f009:**
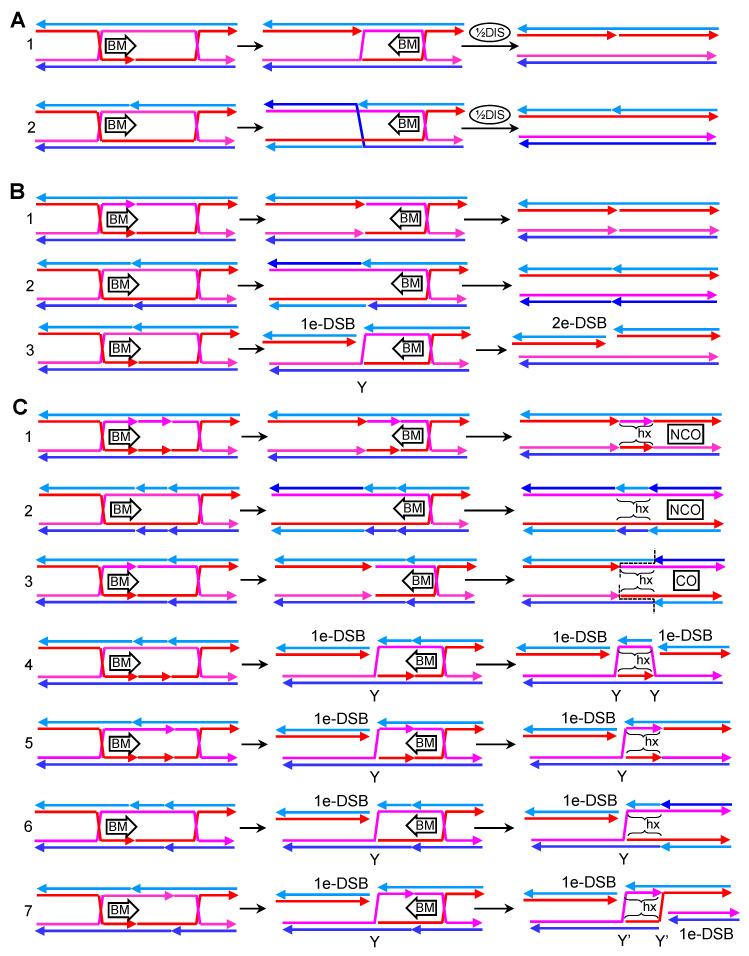
Inferences of metastable nicks during branch migration of a DPE dcHJ. (**A**) Scenarios with one nick: (1) in a crossover strand; (2) in a non-crossover strand. (**B**) Scenarios with two opposing nicks: (1) in the crossover strands; (2) in the non-crossover strands; (3) in one crossover and one non-crossover strand. (**C**) Scenarios with two partners of opposing nicks: (1) all four nicks in the crossover strands; (2) all four nicks in the non-crossover strands; (3) one partner in the crossover strands and the other one in the non-crossover strands; (4) both partners in a crossover/non-crossover arrangement, with only two affected strands; (5) one partner in a crossover/non-crossover arrangement, and the other one in the crossover strands; (6) one partner in a crossover/non-crossover arrangement, and the other one in the non-crossover strands; (7) both partners in a crossover/non-crossover arrangement, with all four strands holding a nick. 1e-DSB, one-ended DSB; 2e-DSB, two-ended DSB; Y and Y’, RF-like structures. Y-Y partner resembles a replication bubble, whereas the Y’-Y’ partner makes a bubble-like JM that connects a CO-to-be chimera. See previous figure legends for other abbreviations and signs.

**Figure 10 genes-11-01498-f010:**
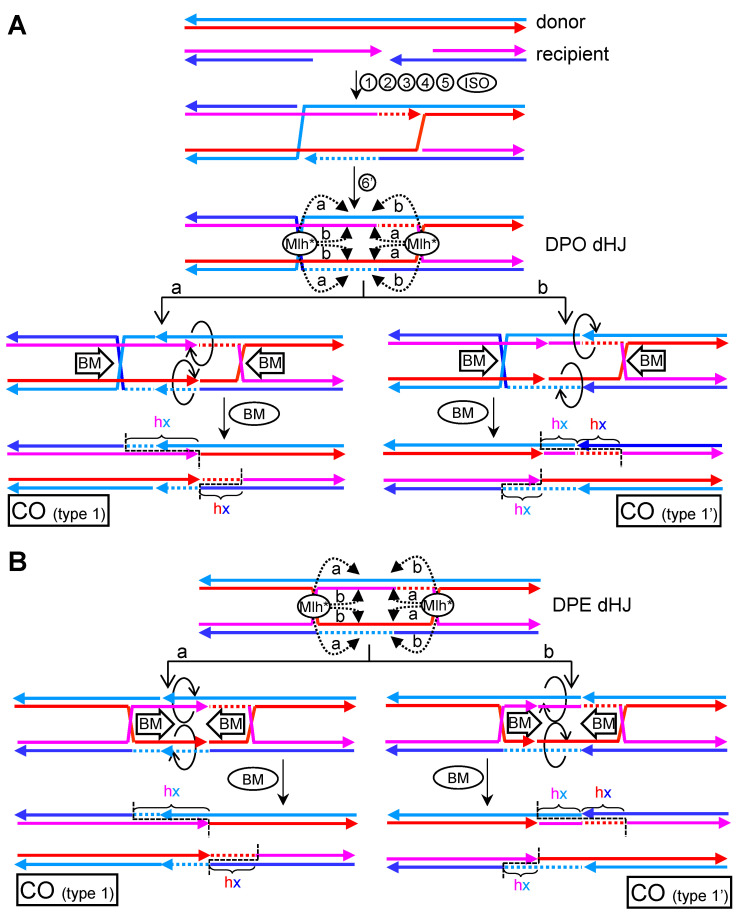
Models for type 1 CO specificity of ZMM meiotic resolution based on Mlh1-Mlh3 making incisions in *trans*. (**A**) Inward nicking by Mlh1-Mlh3 upon a DPO dHJ. Steps (1–5) are like in Figure 7A, and includes the nHJ isomerization (ISO) of Figure 7B. In step 6′, Mlh1-Mlh3 polymerizes inwards from the cHJs/nHJs (cHJs in this example). The main restriction of the model is that incisions in *trans* takes place with strand specificity relative to the nucleating HJ: “a” for crossover strands and “b” for non-crossover strands; “a” is also compatible with targeting first incisions to the newly synthesized strand sections. Branch migration towards the opposing nicks results in the resolution of the cHJs. (**B**) First incisions targeted by the newly synthesized strand section also predict type 1 COs in DPE dcHJ.

**Figure 11 genes-11-01498-f011:**
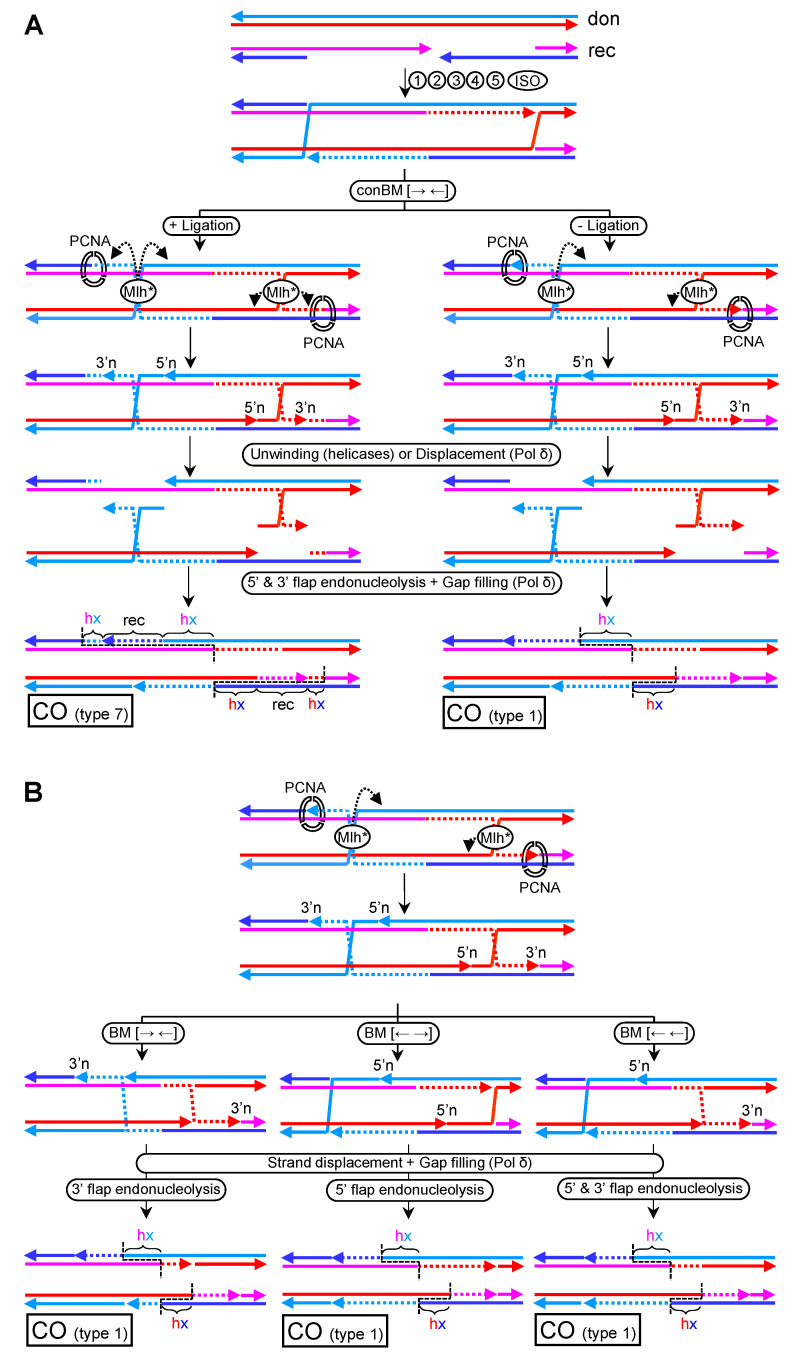
Models for ZMM meiotic resolution based on Mlh1-Mlh3 making incisions in *cis* upon a DPO dHJ. (**A**) Models in which cHJs are resolved from flanking nicks in the crossover strands. Steps 1–5 and ISO are like in Figure 10. Then, convergent branch migration is needed to relocate the newly synthesized tracts within the cross-joins, so that PCNA directs incisions into the crossover strands. Several mechanisms can then process each cHJ from the new nick located 5′ to the cHJ (5′n). To ease visualization, two steps are depicted. Firstly, the unwinding and separation of the portion of nicked crossover strands that stretched across the cHJ, which resolves the cHJ into a 5′-flap and a 3′-flap. This can be achieved by the concerted actions of 3′→5′ and 5′→3′ helicases, or the displacement activity of Pol δ. Secondly, the resulting ssDNA gap is filled in by Pol δ, and the flaps are removed by specialized flap endonucleases. Note that strand displacement and gap filling occur in a single step when Pol δ commences resolution. If a DPO dcHJ is the substrate (left branch; +ligation), a double incision per cHJ is required. Because of the polarity of PCNA, the incision located 3′ to the cHJ (3′n) ought to occur within the newly synthesized track. Thus, the genetic outcome is a reciprocal CO with a hx-recipient-hx tract (type 7). If a DPO dnHJ was the substrate before the converging migration (right branch; -ligation), only the 5′n incision is needed and the mandatory genetic outcome is a type 1 CO, since the 3′n coincides with the end of the HR-driven synthesis by Pol δ. (**B**) Variations of the right branch of A where branch migration towards one of the nicks precedes the action of Pol δ (or helicases; not depicted). Branch migration can be convergent towards the 5′ns (left branch), divergent towards the 3′ns (center) or codirectional (right branch). In all cases, DPO dnHJ are the intermediate metabolites and type 1 COs are the genetic outcome.

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
