# Peer review of "Implications of Metastable Nicks and Nicked Holliday Junctions in Processing Joint Molecules in Mitosis and Meiosis"

_genes, 2020, doi:10.3390/genes11121498_

Round 1

Reviewer 1 Report

In this concept paper, author Felix Machin presents numerous models of processing of joint molecules during meiosis, mitosis, and replication fork stalling. Historically, the canonical repair pathway models currently presented in the literature rely heavily on the assumption that the DNA ends that are involved in intermediates are ligated to form either double Holliday junctions (dHJ) D-loop dissociation.  In this paper, Machin thoroughly presents the canonical DNA double-strand break repair models, focusing on Holliday junction. It contains an excellent review of the existing models and genetic outcomes of each model. He provides novel insight into processing of joint molecules that either retain the nick (i.e., before ligation after second end capture), or during Mlh1-Mlh3 nicking during meiosis.

In short, I find this work to be exceptionally thorough and clear. Machin presents every conceivable possible model and outcome (to my knowledge) in a clear, succinct way, and is well-written. Models are clearly labeled, color coding consistent, which is key to following the paper. This paper could easily be the “go to” body of work of those in the field would visit to explain outcomes of recombination. Many individual papers have attempted to address alternative models that could explain recombination outcomes, but this is the first, to my knowledge, to thoroughly synthesize the canonical models as well as propose alternative models in one manuscript.

I have no major issues with this work that need to be addressed. Only a few minor typos to consider:

  1. In the abstract, Machin stresses the importance of HR in repairing DSBs, protecting from carcinogenesis (i.e., maintain genome stability in somatic cells), and providing genetic diversity. It would be helpful to also include the importance of HR in maintaining proper segregation of homologs during meiosis 1 (as failure to complete HR can lead to non-disjunction).
  2. Line 41 “puts” instead of “put”
  3. Line 64: “established” instead of “stablished”
  4. Line 347: “left” instead of “leave”
  5. Line 404: “Noteworthy, not only can HR deal with DSBs” instead of “Noteworthy, not only HR can deal with DSBs

Reviewer 2 Report

Main concern :

this review is un-necessarily long and complicated, hence very hard to read and follow. It could/should be shortened by half easily, focusing on the key concepts it aims to address. For instance, while nHJ is the main focus of the review, it is first addressed on page 6 only. In addition, merging chapters 2 and 3, for example, seems rather easy since the overlap between the two is huge. Last but not least, it seems mandatory to include the recent findings from the Cejka and the Hunter groups about Mlh1-3 mode of resolution showing the involvement of RFC/PCNA (Nature 2020).

Minor concerns:

- L 56: Keep in mind that in D. melanogaster, MEI-9-ERCC1 endonuclease may be acting as a HJ resolvase.

- L71: “Mus81* is per se a poor resolvase against the canonical HJ (cHJ), yet an excellent nicked 71 HJ (nHJ) resolvase”. Importantly, although this enzyme can exhibit substrate preference for nicked versus fully ligated HJs, this does not mean this enzyme cannot cut fully ligated HJs. In addition, this substrate specificity is altered by post translational modifications as shown by the group of PH Gaillard (Dehé et al 2013) where increased HJ cleavage is seen after phosphorylation of Eme1. Last but not least, in S. pombe, Yen1/GEN1 is absent. If Mus81 was unable to cleave HJs, this would mean that mitotic cHJ could not be resolved in S. pombe, which is unlikely.

- L218: “Most importantly, it is the only four-way substrate of Mus81* [19].” This statement is incorrect.

- It is referred several times to Top1 (ie L195), but without mentioning the way it works. It is simply said it does not work like Top3, but the differences between the two are not presented, while a naïve conception of Top1 action would make it suitable to work on recombination intermediates.

L317: “Besides, and provided that MMR does not quickly deal with the initial hxDNA formed between the two cHJs, RTR dissolution does not cause short-tract GCs as it restores the original complementary strand annealing.” Martini et al 2011 presented evidence that MMR acts early, even on transient hxDNA tracts. This comment also applies to L. 437-441.

- L 810: “The main conclusion of this chapter is that a DPO dHJ is the simplest solution to achieve 100% CO products in the Mlh1-Mlh3 resolution. It is the simplest solution because a DPE dHJ would need alternating strand specificities for each HJ, which appears mechanistically unsightly. Moreover, I present several chief predictions for local hxDNA/GC events that are in agreement with general observations, as I will describe in the next chapter.” This needs to be revisited according to the Cejka and Hunter latest findings about Mlh1-3 mode of resolution showing the involvement of RFC/PCNA.

- Ref 17 is presented as the seminal one about the dissolution pathway, while the group of F. Stahl, at least, presented this model much earlier based on genetic evidence (Gilbertson and Stahl 1996). Same applies for ref 54 used for SDSA while Nassif et al 1994 proposed this model before.

- L 636: “It is generally assumed that JMs arising from a single HR event are processed in a concerted manner”. There is no report about this aspect: the author should acknowledge this fact.

- L 857: should add ref 57 as well with 56, 64-68

- L 858: “of DSB pathways » should be “of DSB repair pathways”

- L 870: “What is needed for type 1 CO is inward nicking of the DPO dHJ with either specificity (crossover or non-crossover strands), as long as at least one nick occurs in the newly-synthetized tract.” It looks to me that similar constraints on a DPE would also generate type 1 CO. So I do not understand the necessity of involving a DPO rather than a DPE here.

A few typos:

L 64: “established” not “stablished”

L 227: “does not participate”

L 450 and others: “the second END capture”, not “the second capture”

L 451 and others: “overhang” not “overhand”

L 492: “are taken” not “are taking”

L 558 and others: “adducts” not “abducts”

L 704: “affects one the non-crossover strands” remove “one”
